# Generalization in Cooperative Multi-Agent Systems

## Abstract

Collective intelligence is a fundamental trait shared by many species that has allowed them to thrive in diverse environmental conditions. From simple organisations in an ant colony to complex systems in human groups, collective intelligence is vital for solving many survival tasks. Such natural systems are flexible to changes in their structure: they generalize well when the abilities or number of agents change, which we call *Combinatorial Generalization* (CG). CG is a highly desirable trait for autonomous systems as it can increase their utility and deployability across a wide range of applications. While recent works addressing specific aspects of CG have shown impressive results on complex domains, they provide no performance guarantees when generalizing to novel situations. In this work, we shed light on the theoretical underpinnings of CG for cooperative multi-agent systems (MAS). Specifically, we study generalization bounds under a linear dependence of the underlying dynamics on the agent capabilities, which can be seen as a generalization of Successor Features to MAS. We then extend the results first for Lipschitz and then arbitrary dependence of rewards on team capabilities. Finally, empirical analysis on various domains using the framework of multi-agent reinforcement learning highlights important desiderata for multi-agent algorithms towards ensuring CG.

## 1 Introduction

Imagine attending a football summer camp. The coach decides to split the participating players into random teams for practice. While each player has different capabilities (e.g., defending, dribbling, speed, and pace), they quickly adapt to the other players in the team to facilitate the common objective of outscoring their opponents. Furthermore, they smoothly adjust to unexpected events such as a player getting hurt and retiring with substitution, which forces them to change their behaviours and adjust their roles. Similarly, they rapidly adjust to changes in team size (as a result of a player being sent off or new players joining the team).

Such adaptations are typically possible for two reasons. First, the players understand each others' *capabilities*, including how a change in capabilities affects the underlying environment and chances of success. Second, players have coordination protocols for adapting to the changes, both explicitly (e.g., communicating the game plan) or implicitly (inferring capabilities from observations, e.g., passing the ball to a player going in for an attack). This phenomenon, which we call *Combinatorial Generalization* (CG), is not specific to football or humans, and organisms in general manifest abilities to adapt in almost every situation requiring team efforts (Crozier et al., 2010; Nouyan et al., 2009; Anderson & McMillan, 2003).

In order to capture specific aspects of CG, recent methods in multi-agent reinforcement learning (MARL) utilize advances in deep learning architectures, such as graph neural networks (Ryu et al., 2020) and attention mechanisms (Iqbal et al., 2021), as well as extensively tuned training regimes, such as a mixture of human and generated data, self-play, and population-based training (Vinyals et al., 2019; OpenAI et al., 2019). While these methods show impressive empirical performance on complex domains, they provide little insight into aspects of when and how much generalization to expect. These are crucial for deploying agents in the real world due to practical considerations like tolerance and minimum expected performance in unseen settings. Additionally, the problem of sample-efficient generalization, already hard for single-agent RL (Mahajan & Tulabandhula, 2017; Du et al., 2020; Ghosh et al., 2021; Malik et al., 2021), is particularly challenging in the multi-agent case. Specifically, even when the underlying task remains the same, agents in MARL typically need to be trained from scratch for different team compositions. Moreover, across similar tasks with similar team compositions, there is a lack of modularity for sharing knowledge to enable quick learning (Wang et al., 2020).

Thus, we posit that a theoretical understanding of generalization in multi-agent systems (MAS) can help address both of the above-mentioned issues: it can provide important performance guarantees for practical deployment and can additionally inform better algorithm design to ensure sample efficiency.

We first highlight the key properties that make CG particularly difficult for MAS:

- **P1:** The capabilities of agents can come from infinite sets, e.g., maximum permissible torque for an agent joint which can take values in a continuous set.

- **P2:** Combinatorial blow-up in the number of possible teams (w.r.t. agent capabilities) given a team size.

- **P3:** The capabilities need to be grounded w.r.t. the dynamics of the environment, ie. the agent needs to infer how the capability affects the long term utility in terms of joint rewards and transitions. This becomes increasingly hard with team size (similar to credit assignment).

- **P4:** Team sizes can vary across different tasks.

- **P5:** Agents need to infer the capabilities of teammates in settings where it is hidden, in a potentially non-stationary environment.

**P2**-**P4** particularly distinguish CG from single-agent generalization, highlighting its combinatorial nature. Furthermore, **P5** requires agents to adapt to changing teammate policies, making the problem harder.

In this work, we analyse multi-agent generalization by modelling the dependence of underlying environment rewards and transitions on agent capabilities. We first look at generalization bounds for the case when the environment dynamics are linear with respect to the agent capabilities. We elucidate how this generalizes the successor feature (SF) framework (Barreto et al., 2016) to the multi-agent case. We provide theoretical bounds for generalization between team compositions, transfer of optimal policy from one team to another and changes to optimal values arising from agent addition and elimination under this framework. Next, we bound the performance gap as a result of an error in estimating the agent capabilities, which covers scenarios such as lossy or inaccurate communication. Furthermore, we provide bounds for optimal value deviation when the dynamics themselves are approximately linear. Finally, we elucidate how the bounds can be extended to Lipschitz rewards (Appendix A.6) and then extend this framework to study arbitrary dependence of rewards on capabilities to shed light on when generalization can be difficult (Appendix A.7). Our results apply to various training and deployment settings in MAS and are agnostic to the type of algorithm used (MARL or other forms of policy search methods). Finally, we empirically analyse popular methods in MARL on tasks of varying difficulty in terms of generalization and discuss important desiderata to be met for better generalization.

## 2 Background and Formulation

### Multi-Agent Reinforcement Learning

We model the cooperative multi-agent task as a decentralized partially observable MDP (Dec-POMDP) (Oliehoek & Amato, 2016). A Dec-POMDP is formally defined as a tuple $G = \langle S, U, P, R, Z, O, n, \rho, \gamma \rangle$. $S$ is the state space of the environment, $\rho$ is the initial state distribution. At each time step $t$, every agent $i \in \mathcal{A} \equiv \{1, ..., n\}$ chooses an action $u^i \in U$ which forms the joint action $\mathbf{u} \in \mathbf{U} \equiv U^n$. $P(s'|s, \mathbf{u}) : S \times \mathbf{U} \times S \to [0, 1]$ is the state transition function. $R(s) : S \to [0, 1]$ is the reward function shared by all agents and $\gamma \in [0, 1)$ is the discount factor. A Dec-POMDP is *partially observable* (Kaelbling et al., 1998): each agent $i$ does not have access to the full state and instead samples observations $z \in Z$ according to observation distribution $O(s, i) : S \times \mathcal{A} \to \mathcal{P}(Z)$. Without loss of generality (WLOG), we assume the state is a represented as a $k$-dimensional feature vector $S \subset [0, 1]^k$ and similarly observations $Z \subset [0, 1]^l$. When the observation function $O$ is identity, the problem becomes a multi-agent MDP (MMDP). Similarly, when the observations are invertible for each agent, so that the observation space is partitioned w.r.t. $S$, i.e., $\forall i \in \mathcal{A}, \forall s_1, s_2 \in S, \forall z_i \in Z, P(z_i|s_1) > 0 \land s_1 \neq s_2 \implies P(z_i|s_2) = 0$, we classify the problem as a multi-agent richly observed MDP (M-ROMDP) (Mahajan et al., 2021). The action-observation history for an agent $i$ is $\tau^i \in T \equiv (Z \times U)^*$. We use $u^{-i}$ to denote the action of all the agents other than $i$ and similarly for the policies $\pi^{-i}$. The value of a policy is defined as $V^\pi = \mathbb{E}_{\pi, \rho} \left[ \sum_{t=0}^\infty \gamma^t R_{\mathcal{T}}(s_t) \right]$, we overload

it to also denote the value function $V^\pi(s) = \mathbb{E}_{\pi,\rho}\left[\sum_{t=0}^\infty \gamma^t R_\mathcal{T}(s_t)|s_0 = s\right]$. Similarly, the joint action-value function given a policy $\pi$ is defined as: $Q^\pi(s_t, \mathbf{u}_t) = \mathbb{E}_\pi\left[\sum_{k=0}^\infty \gamma^k R(s_{t+k})|s_t, \mathbf{u}_t\right]$. The goal is to find the optimal policy $\pi^*$ corresponding to the optimal value function $V^*$.

### MARL with Agent Capabilities

We now extend the MARL problem setting for generalisation where agents can have different capabilities. To this end, we assume that each agent in the task can be characterised by a $d$-dimensional *capability vector $c \in \mathcal{C}$*, which governs its contribution to rewards and transition dynamics (and thus its policy/behaviour denoted as $\pi^i(\ .\ ;c)$). Without loss of generality, we assume $\mathcal{C} \subseteq \Delta_{d-1}$ (the $d - 1$ dimensional simplex). Intuitively, an agent's capability reflects the abilities of an agent along various properties that may be important for solving the collective task (e.g., an agent's speed, health recovery, and accuracy). We next assume an unknown probability distribution $\mathcal{M} : \mathcal{C}^n \to \mathbb{R}^+$ with support $Sup(\mathcal{M})$ over a subset of the joint capability space $\mathcal{C}^n$. Any $\mathcal{T}$ sampled from $\mathcal{M}$ can be seen as a tuple of capability vectors $\mathcal{T} = (c_i)_{i=1}^n$, one for each agent in the team. We augment the Dec-POMDP with $\mathcal{T}$: $G = \langle S, U, P_\mathcal{T}, R_\mathcal{T}, Z, O, n, \rho, \gamma, \mathcal{T} \rangle$ and call it a *variation* for the MARL setting [1]. Thus $\mathcal{T}$ defines the rewards and transition dynamics of the underlying MMDP (ie. $R_\mathcal{T}(s) = \langle f(\mathcal{T}) \cdot s \rangle$ where $\langle \cdot \rangle$ is the dot product[2] and $f : \mathcal{C}^n \to \mathbb{R}^k$ and similarly for transitions). Our goal is then to find algorithms, which when trained on a small number of *variations* sampled from $\mathcal{M} : \{\mathcal{T}^j\}_{j=1}^M$, generalise well to unseen team variations in $\mathcal{M}$. i.e., we want to maximise the expected value over the team variation distribution,

$$\max_\pi \ \mathbb{E}_{\mathcal{T} \sim \mathcal{M}}\left[\mathbb{E}_{\pi(\cdot;\mathcal{T}),P_\mathcal{T},\rho}\left[\sum_{t=0}^\infty \gamma^t R_\mathcal{T}(s_t)\right]\right], \tag{1}$$

where $\pi = \{\pi^i\}_{i=1}^n$ is a group of $n$ agents. The challenge here arises because of two main factors. First, the agents do not have any prior knowledge about what these capability vectors mean, and are thus required to learn their semantics (also called grounding). Second, in the setting where the agents cannot observe the capability vectors (including possibly their own), they have to infer and learn protocols for sharing them with each other in order to generalize in a zero-shot setting.

### Successor Features

Thus successor features (SF) framework (Dayan, 1993; Barreto et al., 2016; 2018; 2020) assumes that the rewards in an MDP can be decomposed as $r(s) = \boldsymbol{\phi}(s)^\top \mathbf{w}$, where $\boldsymbol{\phi}(s) \in \mathbb{R}^d$ are features of $s$ and $\mathbf{w} \in \mathbb{R}^d$ are weights[3]. When no assumption is made about $\boldsymbol{\phi}(s)$, any reward function can be recovered using this representation. The value function then follows

$$\begin{aligned}
V^\pi(s) &= \mathbb{E}^\pi\left[r_{t+1} + \gamma r_{t+2} + ... \mid S_t = s\right] \\
&= \mathbb{E}^\pi\left[\boldsymbol{\phi}_{t+1}^\top \mathbf{w} + \gamma \boldsymbol{\phi}_{t+2}^\top \mathbf{w} + ... \mid S_t = s\right] \\
&= \boldsymbol{\psi}^\pi(s)^\top \mathbf{w}.
\end{aligned}$$

Here $\boldsymbol{\psi}^\pi(s)$ is called the *successor feature* of $s$ under policy $\pi$. The $i$th component of SF $\boldsymbol{\psi}^\pi(s)$ provides the expected discounted sum of $\phi_i$ when following policy $\pi$ from $s$.

## 3 Analysis

As mentioned before, we are interested in understanding how the long term joint-utility of a cooperative group changes with changes happening in the group. Our analysis here focuses on the generalisation properties w.r.t. $\mathcal{M}$. We focus on the case of MMDPs for ease of exposition, but similar results for the more general cases can be obtained by suitable assumptions for identifiability of the state (e.g., M-ROMDP in Mahajan et al. (2021)). Our results are applicable irrespective of whether agents can observe the capabilities. They are also agnostic to the training and deployment regimes (e.g., centralized or decentralized) and the algorithm being used to find the policy. **All the proofs can be found in Appendix A.** For the analysis we assume

---

[1]Agent capabilities can also be interpreted as the contexts, see Hallak et al. (2015)

[2]Note that this is still the most general form as states can be encoded as one-hot vectors, see Barreto et al. (2016).

[3]Similar formulations hold WLOG for $\phi$(s,a),$\phi$(s,a,s')

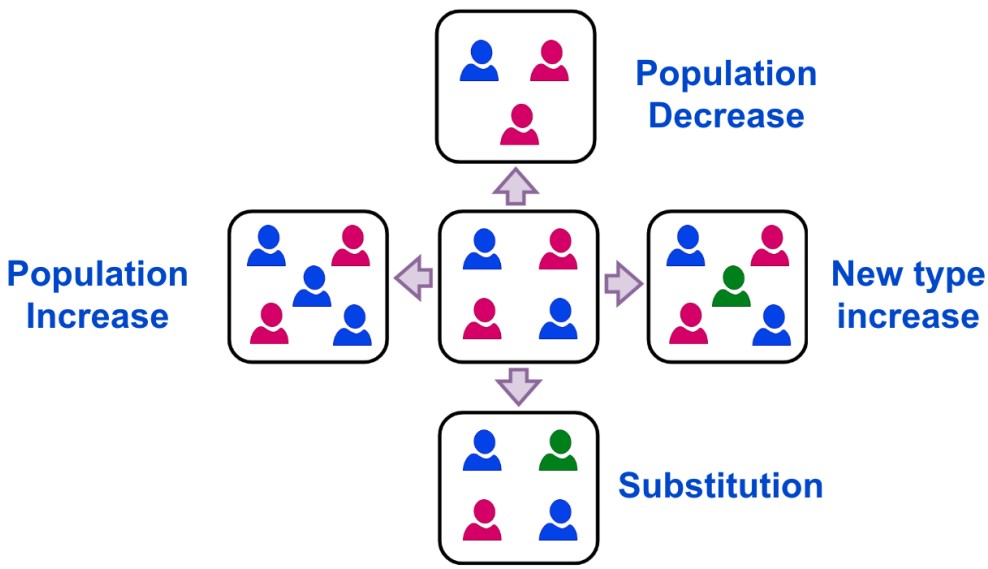

Figure 1: Combinatorial Generalization in MAS, various settings.

that the rewards and transitions depend linearly on the agents capabilities $c_i$ :

$$R_{\mathcal{T}}(s) = \sum_{i=1}^{n} a_i \langle c_i \cdot W_R s \rangle \tag{2}$$

$$P_{\mathcal{T}}(s'|s, \mathbf{u}) = \sum_{i=1}^{n} a_i \langle c_i \cdot W_P(s', \mathbf{u}, s) \rangle \tag{3}$$

where $W_R \in \mathbb{R}^{dk}$ is the reward kernel of the MMDP and defines the dependence of the rewards on each capability component. Similarly in Eq. (3), $W_P : S \times \mathbf{U} \times S \times \{1..d\} \to [0, 1]$ defines the transition kernel of the MMDP so that $P_j(\cdot|s, \mathbf{u}) \triangleq W_P(\cdot, \mathbf{u}, s, j) \in \Delta_{|S|-1}, j \in \{1..d\}$ give the next state distribution as directed by the $j^{th}$ component of the capability and agent $i$'s propensity (unweighted) to make the state transition to $s'$ is given by $\left\langle c_i \cdot \left[ P_1(s'|s, \mathbf{u}) \dots P_d(s'|s, \mathbf{u}) \right] \right\rangle = \langle c_i \cdot W_P(s', \mathbf{u}, s) \rangle$. Finally $(a_i)_{i=1}^{n} \in \Delta_{n-1}$ are the *influence weights* of agents which quantify the influence of agent $i$ in determining the rewards and transitions. Under the linear setting, given a policy $\pi$ and capabilities $\mathcal{T}$ we have that value function satisfies $V_{\mathcal{T}}^{\pi} = \sum_{i=1}^{n} a_i \langle c_i \cdot W_R \mu_{\mathcal{T}}^{\pi} \rangle$ where $\mu_{\mathcal{T}}^{\pi} = \mathbb{E}_{\rho, P_{\mathcal{T}}, \pi}[\gamma^t s_t]$ are the expected discounted state features and similarly for a given state $s$, $V_{\mathcal{T}}^{\pi}(s) = \sum_{i=1}^{n} a_i \langle c_i^T W_R \cdot \mu_{\mathcal{T}}^{\pi}(s) \rangle$ where $\mu_{\mathcal{T}}^{\pi}(s) = \mathbb{E}_{P_{\mathcal{T}}, \pi}[\gamma^t s_t|s_0 = s]$. The linear formulation for dynamics generalizes the successor feature (Barreto et al., 2016) formulation to the MAS setting, this can be seen by noting that when the dependence of transition dynamics on capabilities is dropped (Eq. (3)) and only single agent is considered (by considering a one-hot $a$), we get the successor feature formulation with capability of the nonzero $a_i$ interpreted as the task weight in Barreto et al. (2016)(see Section 2). We now present the first result concerning the difference between the optimal values of two different team compositions:

**Theorem 1** (Generalisation between team compositions)**.** *Let team compositions* $\mathcal{T}^x, \mathcal{T}^y \in \mathcal{C}^n$ *with influence weights* $a^x, a^y \in \Delta_{n-1}$, $s_{max} = \max_s ||W_R s||_1$, $V_{mid} = \frac{1}{2} \max_s V_{\mathcal{T}^y}^*(s)$, *then*[4]:

$$|V_{\mathcal{T}^x}^* - V_{\mathcal{T}^y}^*| \leq \frac{s_{max} + \gamma d V_{mid}}{\gamma(1 - \gamma)} \Psi, \text{ where}$$

$$\Psi = \left[ |\sum_i a_i^x (\mathcal{T}_i^x - \mathcal{T}_i^y)|_{\infty} + |\sum_i (a_i^x - a_i^y) \mathcal{T}_i^y|_{\infty} \right] \tag{4}$$

---

[4]for $\gamma \in (0, \frac{\sqrt{5}-1}{2})$ we can replace $\frac{1}{\gamma(1-\gamma)}$ by $\frac{1+\gamma}{1-\gamma}$

Theorem 1 gives an interesting decomposition of an upper bound to the difference of the optimal values between the two team compositions. The first terms in the square brackets on the RHS denote contributions arising purely from substituting the old capacities with the new one. The second term denotes the contribution arising from a change in how much influence the agents have over the dynamics of the MMDP.

**Corollary 1.1** (Change in optimal value as a result of agent substitution). *Let $\mathcal{T} \in \mathcal{C}^n$ be a team composition with influence weights $a \in \Delta_{n-1}$. If agent $i$ is substituted with $i'$ keeping $a_i$ unchanged such that $|\mathcal{T}_{i'} - \mathcal{T}_i|_\infty \leq \epsilon_C$ then the new team ($\mathcal{T}'$) optimal value follows:*

$$|V_{\mathcal{T}'}^* - V_{\mathcal{T}}^*| \leq \frac{(s_{max} + \gamma dV_{mid})a_i\epsilon_C}{\gamma(1-\gamma)}$$

We define an important policy concept which captures the absolute optimality for an oracle with access to the capabilities. For the ease of exposition we consider fixed influence weights $a$ and define a metric on the joint capability space as $d_a(\mathcal{T}^x, \mathcal{T}^y) = |\sum_i a_i(\mathcal{T}_i^x - \mathcal{T}_i^y)|_\infty$. We similarly generalize this metric to distances between sets by taking the infimum of the distances between pairs of points in the cross product $d_a(\mathcal{M}_x, \mathcal{M}_y) \triangleq \inf_{\mathcal{T}^x \in \mathcal{M}_x, \mathcal{T}^y \in \mathcal{M}_y} d_a(\mathcal{T}^x, \mathcal{T}^y)$.

**Definition 1** (Absolute Oracle). *Let $\pi_{\mathcal{M}}^*$ be the oracle policy which optimizes Eq. (1) ie. $\pi_{\mathcal{M}}^*$ is the multiplexer policy which given a team composition $\mathcal{T}$ behaves identically to the optimal policy for $\mathcal{T}^j$ where $\mathcal{T}^j \in \arg\min_{\mathcal{T}^l \in Sup(\mathcal{M})} d_a(\mathcal{T}^l, \mathcal{T})$.*

We now answer the question of what happens when agents are trained on specific capabilities but the learnt policy is used on potentially unseen capabilities (this could occur, e.g., due to changes in hardware components).

**Theorem 2** (Transfer of optimal policy). *Let $\mathcal{T}^x, \mathcal{T}^y \in \mathcal{C}^n$, $a^x, a^y \in \Delta_{n-1}$, $s_{max} = \max_s ||W_R s||_1$, $V_{mid} = \frac{1}{2}\max_s V_{\mathcal{T}^y}^*(s)$. Let $\pi_y^*$ be the optimal policy for the team composed of agents with capabilities $\mathcal{T}^y$ and influence weights $a^y$. Then:*

$$V_{\mathcal{T}^x}^* - V_{\mathcal{T}^x}^{\pi_y^*} \leq 2\frac{s_{max} + \gamma dV_{mid}}{\gamma(1-\gamma)}\Psi,$$

*where $\Psi$ is defined as in Eq. (4).*

**Corollary 2.1** (Out of distribution performance). *Let $\mathcal{T} \notin Sup(\mathcal{M})$ be an out of distribution task, we then have that the performance of the absolute oracle policy on $\mathcal{T}$ satisfies:*

$$V_{\mathcal{T}}^* - V_{\mathcal{T}}^{\pi_{\mathcal{M}}^*} \leq 2\frac{s_{max} + \gamma dV_{mid}}{\gamma(1-\gamma)}d_a(\mathcal{T}, Sup(\mathcal{M})),$$

We now address the scenarios when the team population changes.

**Theorem 3** (Population decrease bound). *For the team composition $\mathcal{T} \in \mathcal{C}^n$ with influence weights $a \in \Delta_{n-1}$. If agent $n$ is eliminated followed by a renormalization of influence weights, we have that for the remaining team ($\mathcal{T}^- \triangleq (\mathcal{T})_{i=1}^{n-1}$):*

$$|V_{\mathcal{T}^-}^* - V_{\mathcal{T}}^*| \leq \frac{a_n(s_{max} + \gamma dV_{mid})}{\gamma(1-\gamma)}\left|\sum_{i=1}^{n-1}\frac{a_i\mathcal{T}_i}{1-a_n} - \mathcal{T}_n\right|_\infty.$$

The special case when $\sum_{i=1}^{n-1}\frac{a_i\mathcal{T}_i}{1-a_n} = \mathcal{T}_n$ for the linear dynamics formulation when an agent $n$ can in principle be rendered redundant if the rest of the agents in the team can effectively provide a perfect substitute. In fact, this holds true as long as capacity $\mathcal{T}_n$ can be formed from a convex combination of the capabilities $\mathcal{T}_i, i \in \{1..n-1\}$. The latter case however requires using the corresponding convex coefficients instead of re-1normalization. A similar bound can be easily constructed for reusing the policy after an agent eliminated to give the corresponding transfer bound along the lines of Theorem 2.

**Corollary 3.1** (Population increase bound). *For the team composition $\mathcal{T} \in \mathcal{C}^n$ with influence weights $a \in \Delta_{n-1}$. If agent $n+1$ is added with capability $\mathcal{T}_{n+1}$ and weight $a_{n+1}$ (other weights scaled down by $\lambda = 1 - a_{n+1}$) we have that for the new team ($\mathcal{T}^+ \triangleq (\mathcal{T}_1..\mathcal{T}_n, \mathcal{T}_{n+1})$):*

$$|V_{\mathcal{T}^+}^* - V_{\mathcal{T}}^*| \leq \frac{a_{n+1}(s_{max} + \gamma dV_{mid})}{\gamma(1-\gamma)}\Big|\sum_{i=1}^n a_i \mathcal{T}_i - \mathcal{T}_{n+1}\Big|_{\infty}.$$

We next extend the generalization bound Theorem 1 to include the scenario where the reward and the transition dynamics are not exactly linear but are approximately linear with deviation $\hat{\epsilon}_R, \hat{\epsilon}_P$ respectively.

**Theorem 4** (Approximate $\hat{\epsilon}_R, \hat{\epsilon}_P$ dynamics). *Let $\mathcal{T}^x, \mathcal{T}^y \in \mathcal{C}^n$, $a^x, a^y \in \Delta_{n-1}$ and the dynamics be only approximately linear so that $|R_{\mathcal{T}}(s) - \sum_{i=1}^n a_i \langle c_i \cdot W_R s \rangle| \leq \hat{\epsilon}_R$ and $|P_{\mathcal{T}}(s'|s, \mathbf{u}) - \sum_{i=1}^n a_i \langle c_i \cdot W_P(s', s, \mathbf{u})\rangle| \leq \hat{\epsilon}_P$. Then:*

$$|V_{\mathcal{T}^x}^* - V_{\mathcal{T}^y}^*| \leq \frac{s_{max} + \gamma dV_{mid}}{\gamma(1-\gamma)}\Psi + \frac{2(\hat{\epsilon}_R + \gamma\hat{\epsilon}_P V_{mid})}{\gamma(1-\gamma)},$$

*where $\Psi$ is defined as in Eq.* (4).

The other bounds for transfer and population change can similarly be obtained for the approximate dynamics case.

We now consider the scenario when the capabilities are not directly observed but inferred using an approximator which in turn introduces some errors in their estimation (this could happen due to noise in observations, inaccurate implicit or explicit communication protocols, etc.).

**Theorem 5** (Error from estimation of capabilities). *For the team composition $\mathcal{T} \in \mathcal{C}^n$ with influence weights $a \in \Delta_{n-1}$. If the agent capabilities are inaccurately inferred as $\hat{\mathcal{T}}$ with $\max_i |\mathcal{T}_i - \hat{\mathcal{T}}_i|_{\infty} \leq \epsilon_{\mathcal{T}}$ and agents learn the inexact policy $\hat{\pi}^*$ then:*

$$|V_{\mathcal{T}}^* - V_{\mathcal{T}}^{\hat{\pi}^*}| \leq \frac{2\epsilon_{\mathcal{T}}(s_{max} + \gamma dV_{mid})}{\gamma(1-\gamma)},$$

*where $V_{mid} = \frac{1}{2}\max_s V_{\mathcal{T}}^*(s)$.*

All the above results can be easily extended to the setting where rewards $R_{\mathcal{T}}(s) = \langle f(\mathcal{T}) \cdot W_R s \rangle$, $f(\mathcal{T})$ is not linear in capabilities as in Eq. (2) but is Lipschitz with coefficient $L_i$ for $i \in \mathcal{A}$. Note that any non-linear dependence where capabilities belong to a bounded space satisfies Lipschitz boundedness. This is an important extension because it helps us model more complex, non-linear dependence of the underlying dynamics on the agent capabilities. For example, Theorem 1 becomes:

**Theorem 6.** *For rewards $L_i$ Lipschitz in the capabilities with respect to $|\cdot|_{\infty}$ norm, the difference in optimal values between team compositions $\mathcal{T}^x, \mathcal{T}^y$ satisfy:*

$$|V_{\mathcal{T}^x}^* - V_{\mathcal{T}^y}^*| \leq \frac{s_{max}\sum_{i=1}^n L_i|\mathcal{T}_i^x - \mathcal{T}_i^y|_{\infty}}{\gamma(1-\gamma)}.$$

See Appendix A.6 for the proof, which also provides a method for extending the other results in a similar fashion. Thus our results can easily be extended to the settings where the dependence on capabilities is non-linear.

We next take a closer look at the case of general, non-linear dependence of $f$ on $\mathcal{T}$ (as is common for dense capability embeddings) more details for which can be found in Appendix A.7. We also present an insight as to why generalization becomes harder in this setting. To study the case of general, non-linear dependence of rewards on the capabilities in the most general form, we introduce the notion of $(\alpha, k)$-rewards where $\alpha \geq 0, k \in \mathbb{N}$.

$$R_{\mathcal{T}}(s) = \Big\langle \sum_{k_i \in \mathbb{N}, \sum k_i \leq k} a_{k_1..k_n}\Pi_{i=1}^n c_i^{k_i} \cdot W_R s \Big\rangle \tag{5}$$

where $\mathbb{N}$ are non negative integers, $|a_{k_1..k_n}| \leq \alpha$ and $c_i^k{}_i$ represents element-wise exponentiation. Rewards in Eq. (2) can be seen as a special case belonging to Eq. (5) the choice $\alpha, k = 1$. Similarly the union $\cup_{\alpha \geq 0, k \in \mathbb{N}}(\alpha, k)$-rewards cover all possible reward dependencies on capabilities. We have further relaxed the assumption of influence weights belonging to a simplex here and replaced it with individual bounds on the power series coefficients here. We next see that for this scenario, even a small change in the capability of a single agent can shift the rewards massively. Let the capability of agent $i$ be changed from $\mathcal{T}_i$ to $\mathcal{T}_{i'}$ such that $|\mathcal{T}_i - \mathcal{T}_{i'}|_\infty \leq \delta$. Then we have

**Lemma 1.** *For substitution $\mathcal{T}_i$ to $\mathcal{T}_{i'}$ such that $|\mathcal{T}_i - \mathcal{T}_{i'}|_\infty \leq \delta$ under the $(\alpha, k)$-rewards setting we have that*

$$
\begin{aligned}
\epsilon_R &= \max_{s \in S} \left| \langle f(\mathcal{T}^x) \cdot W_R s \rangle - \langle f(\mathcal{T}^y) \cdot W_R s \rangle \right| \\
&= \max_{s \in S} \left| \left\langle \sum_{k_i \in \mathbb{N}, \sum k_i \leq k} a_{k_1..k_n} \Pi_{j \neq i} \mathcal{T}_j^{k_j} (\mathcal{T}_i^{k_i} - \mathcal{T}_{i'}^{k_i}) \cdot W_R s \right\rangle \right| \\
&\leq \max_{s \in S} \left| \sum_{k_i \in \mathbb{N}, \sum k_i \leq k} a_{k_1..k_n} \Pi_{j \neq i} \mathcal{T}_j^{k_j} (\mathcal{T}_i^{k_i} - \mathcal{T}_{i'}^{k_i}) \right|_\infty \left| W_R s \right|_1 \\
&\leq \alpha s_{max} \sum_{j=0}^{k} \sum_{l=1}^{j} \binom{l}{j} l |\mathcal{T}_i^{k_i} - \mathcal{T}_{i'}^{k_i}|_\infty \\
&\leq \alpha \delta s_{max} \sum_{j=0}^{k} j 2^{j-1} = \mathcal{O}(\alpha \delta s_{max} k 2^k)
\end{aligned}
$$

*The above gives us:*

$$
|V_{\mathcal{T}^x}^* - V_{\mathcal{T}^y}^*| \leq \frac{\mathcal{O}(\alpha \delta s_{max} k 2^k)}{\gamma(1-\gamma)}
$$

*where $\mathcal{T}^x, \mathcal{T}^y$ are the joint capabilities before and after agent $i$ capability is changed respectively and $\mathcal{O}(\cdot)$ denotes the order of the term.*

The above suggests that even a small change in the capability of an agent can cause the rewards to change by a lot, hence it is natural to expect that generalization becomes harder as the problem start showing the needle in the haystack phenomenon where only the *right combination* of capabilities gives a large optimal value.

We provide experiments elucidating the bounds stated above in Section 5.1.

## 4 Experimental Setup

We evaluate the ability of existing MARL algorithms to generalize to novel settings where the capabilities of teammates change during the training. We are interested in evaluating the gap between settings encountered during training and held-out agent configurations reserved for testing. Furthermore, we aim to study how well algorithms ground privileged information about teammate capabilities and use that during unseen settings at test time. Lastly, we evaluate the bounds derived in Section 3 on a simple multi-agent problem. Code for the setup is provided in supplementary material.

### 4.1 Environments

We first describe the motivation for the choice of the experimental domains we use below: The Fruit Forage follows the linear dependence in Eq. (2),Eq. (3) and is used to empirically validate the various bounds in Section 3 since the optimal policies can be manually computed for this domain. The Predator Prey and StarCraft II environments represent more challenging scenarios of non-linear dependence of the underlying reward and transitions on the agent capabilities discussed in Section 3 (Theorem 6, Lemma 1).

### 4.1.1 Fruit Forage

We use the fruit forage task on a grid world to empirically demonstrate the generalisation bounds in Section 3. On a $8 \times 8$ grid world we have $n$ agents and $d$ types of fruit trees. For each agent $i$, $\mathcal{T}_i(j), j \in \{1..d\}$

represents the utility of fruit $j$ for agent $i$. The state vector is appended with the $d$ dimensional binary vector representing whether each of the tree types has foraged at a given time step. The details for the team compositions can be found in Appendix B.1.1.

### 4.1.2 Predator Prey

We consider the grid-world version of the multi-agent Predator Prey task where 4 agents have to hunt 4 prey in an $8 \times 8$ grid. Here, both predators and prey have certain capabilities. Specifically, each predator has a parameter describing the hit point damage it can cause the prey. Similarly, the prey comes with variations in health. For example, a prey with a capability of 5 can only be caught if the total capability of agents taking the capture action simultaneously on it have capabilities $\geq 5$ (such as $[1,1,1,2]$), otherwise, the whole team receives a penalty $p$. Here, we test for generalization to novel team composition where test tasks contain a team composition which has not been encountered during training (PP Unseen Team in Figure 4), and additionally test tasks where novel team compositions can also have agent types with capabilities not encountered during training (PP Unseen Team, Agent in Figure 4). More details are provided in the Appendix B.1.2.

### 4.1.3 StarCraft II

To assess the generalization capabilities of modern MARL approaches, we make use of a modified version of StarCraft II unit micromanagement tasks of the SMAC benchmark (Samvelyan et al., 2019). Particularly, we consider novel scenarios featuring three unit types from each race of the game where the team composition changes during training and testing, unlike standard SMAC which is static. The opponent's team is always identical to the ally team which ensures that we can directly compare the joint policy with the game AI policy. In the simple cases (`10_Protoss`, `10_Zerg`, and `10_Terran`), agents are trained on various team formations of 10 units that feature all combinations of one, two, and all three unit types, and is later tested on held out team formations.

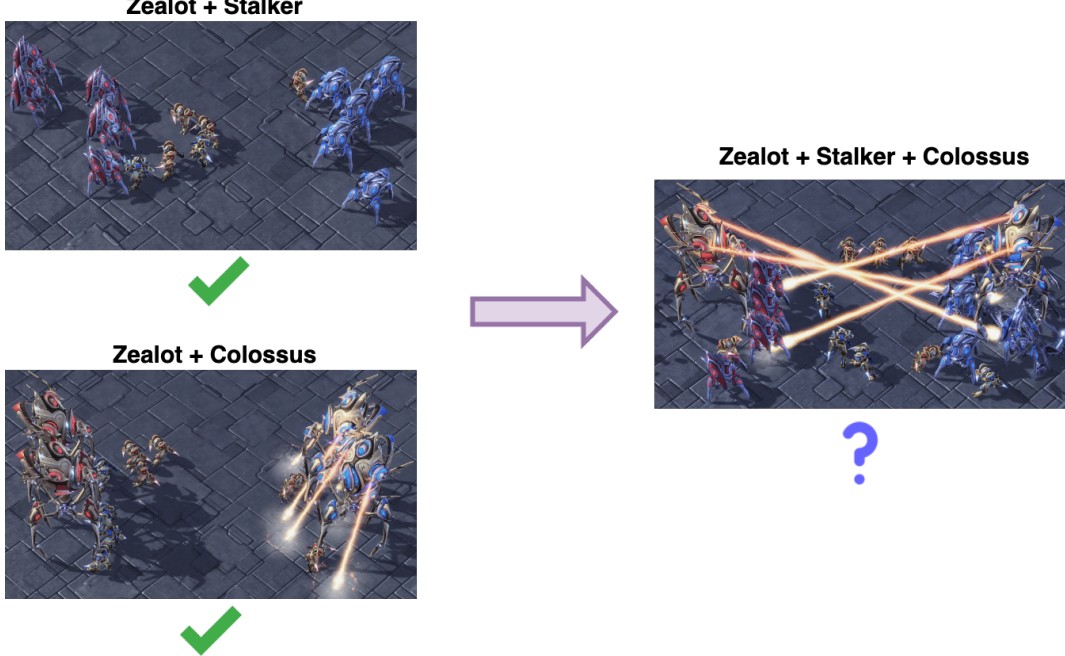

Figure 2: Three episodes from the `10_Protoss_Hard` task (a) Top-left featuring only Zealot and Stalkers during training. (b) Bottom-left featuring only Zealot and Colossus during training. (c) Right: A held-out episode featuring Zealot, Stalker, and Colossus encountered during testing.

In the hard cases (`10_Protoss_Hard`, `10_Zerg_Hard`, and `10_Terran_Hard`), agents are exposed to various team formations including two unit types during training. During testing, however, the agents encounter held-out scenarios featuring scenarios with using all three unit types (see Appendix B.1.3 for more details).

Fig. 2 illustrates three episodes from the `10_Protoss_Hard` environment. In these tasks, agent capabilities are described as a one-hot encoding of agent types.

To test performance on continuously varying capabilities, we also use variants of the environment where either the health or attack accuracy of certain units are reduced. We randomize these configurations for the allied units during training and later test on held-out team configurations. We evaluate baselines on the `3m`, `2s3z`, `8m` scenarios from the original benchmark with these modifications. The varying team size also helps understand how grounding the capabilities becomes harder as team size increases. Here agent capabilities are described as their accuracy or health coefficients. Further details are provided in the Appendix B.1.3.

### 4.2 Baselines

Our empirical evaluation is based on various types of MARL algorithms. We use two popular value-based approaches, QMIX (Rashid et al., 2020) and VDN (Sunehag et al., 2017) that train fully decentralized policies in a centralized fashion. We also use the policy gradient method PPO (Schulman et al., 2017) that has recently shown good results on various MARL domains, both with decentralised (Independent PPO) (de Witt et al., 2020) and centralised critics (MAPPO) (Yu et al., 2021). We assess the performance of all baselines when the information about teammates capabilities are provided as observation (denoted with a 'C' in parentheses) and when it is not, these two variations denote the extreme situations about the teammate capability knowledge. To learn good generalizable policies in the situation where agents can observe the teammate capabilities (dashed lines), the agents must learn to ground the capabilities they observe, this case covers challenges P1-P4 in Section 1. Whereas, for the case when they do not observe the capabilities (solid lines), learning is harder as the agents must also learn to infer the teammate capabilities in a non stationary environment as all the agent policies are changing, this further adds challenge P5 for the agents. Note that the performance in situations which allow explicit communication for informing others about capabilities must lie between the above two extremes. From implementation standpoint, the architecture for the two baseline variations is exactly the same with the teammate capabilities masked for the solid plot lines. **The evaluation procedure, architectures and training details are presented in Appendix B.2.**

## 5 Results and Discussion

### 5.1 Generalization Bounds

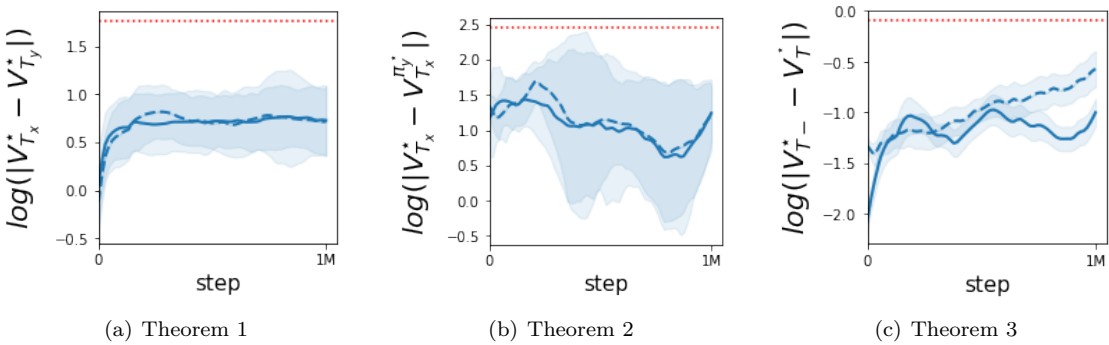

(a) Theorem 1          (b) Theorem 2          (c) Theorem 3

Figure 3: Evaluating the bounds for QMIX on Fruit Forage domain. Dashed blue line indicates the setting where agent capabilities are observable. The red dotted line indicates the corresponding upper bound for each theorem.

Fig. 3 provides empirical evaluation of bounds presented in Section 3 in the Fruit Forage domain. We present the plots for training the agents for one million steps of training using QMIX. Fig. 3(a) shows that the policies in both the domains converge quickly leading to a stable difference in performance thus comfortably satisfying Theorem 1. Fig. 3(b) shows the gap between optimal and transferred policy and reveals interesting variations as training proceeds (we posit this happens because the transferred policy becomes steadily specialized thus getting less useful for the target task); the bound in Theorem 2 gives a tight fit despite these variations. Finally, we see similarly good fit for the agent elimination scenario in Theorem 3 in Fig. 3(c).

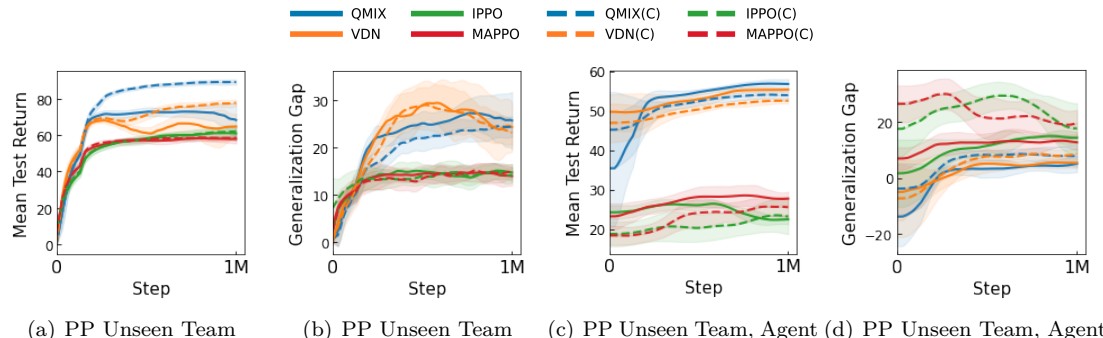

Figure 4: Experimental results for the Predator Prey domain. Standard deviation is shaded.

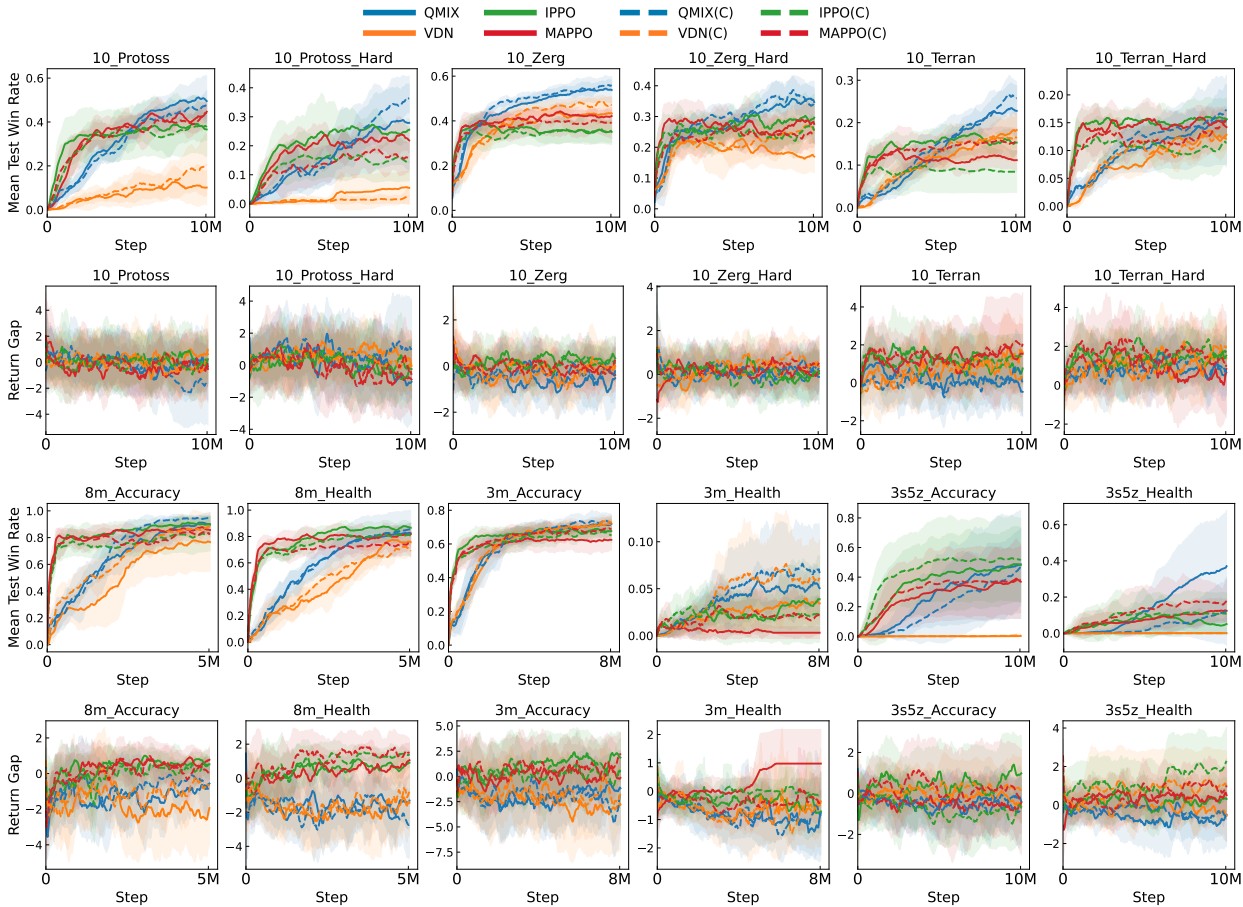

Figure 5: Experimental results on the SMAC benchmark. Standard deviation is shaded. Rows show win rates and generalization gaps.

## 5.2 Utilizing Information of Agent Capabilities

Fig. 4 presents the results of the baselines on Predator Prey domain. Fig. 4(a) shows that providing additional information on agent capabilities improves the test-time performance of the baselines with the maximal effect seen on QMIX and VDN. Furthermore, Fig. 4(b) shows that when capabilities are observable to the agents, baselines are able to generalize to new team compositions, thus successfully grounding the additional information (The only difference between dashed and solid line is observing the teammate capabilities and other details like network architecture is identical. Therefore, the performance difference is solely attributable

to access to teammate capabilities. The only way to use capabilities is to "ground" them ie. meaningfully interpret them through the agent neural net.) This hypothesis is additionally supported by the fact that knowing agent capabilities results in a lower generalization gap. Finally, the gap between the settings with known vs. unknown capabilities (dashed vs. solid) indicates that agents have likely not come up with any appropriate protocol to communicate their capabilities during test time. Furthermore, the PPO variants do not perform as well as the value-based approaches. Therefore, their low generalization gap Fig. 4(b) is unlikely representative of good grounding of capability. We posit that this is just because PPO agents are ignoring the privileged information when available.

For a harder scenario, where both new team composition and agent types appear during evaluation, Fig. 4(c) shows that the situation is reversed from the previous setting as the agents that do not have access to each other's capabilities now perform slightly better. This is strongly indicative of insufficient grounding of the privileged information given to them, which highlights the need for better grounding mechanisms to obtain CG. We see a similar pattern on generalization gap in Fig. 4(d) where privileged information hurts the performance and is likely perceived as observation noise, this again follows because every detail except for teammate capability, like architecture etc. is common between the dashed and solid lines.

On the more challenging domain of StarCraft, we see that for easier capability variations of health and accuracy (as they are continuous and more readily usable for an agent's immediate actions), knowing the capabilities is advantageous to the agent during test time. Moreover, the relative gains of knowing the privileged information go down as the task difficulty increases. The accuracy variations tend to be easier as typical joint policies like 'focus fire' where a group of units to attack a single target together remain unchanged. Moreover, health variations on smaller teams make the task much harder than on bigger teams due to relative loss in team hit points. In this regard, `8m, 3s5z` accuracy versions show good grounding and generalization. This changes as tasks get harder. On the harder tasks that involve swapping unit types within `Protoss, Zerg, Terran` races, we observe that knowing the capabilities of other agents gives little advantage. This is especially noticeable on the `Hard` versions where all unit types are never within a single team during training. Furthermore, with win-rate performances on these maps being low, we hypothesise that the agents do not successfully utilize the capability information. Thus, it is highly unlikely that they learn any meaningful communication protocols for exchanging capability information. **For full StarCraft II results, including `8m_vs_9m` & `10m_vs_11m` scenarios, see Appendix C**.

Compared to the relatively simple Predator Prey task, generalization in StarCraft proved to be more difficult for the baselines. Although static versions of SMAC environments are comfortably solved by them (Rashid et al., 2020; de Witt et al., 2020; Yu et al., 2021), changing unit formations or unit health/accuracy makes the tasks significantly difficult, even for configurations seen during the training. We therefore conclude that in such challenging high-dimensional environments, simply providing agent capabilities as input to agents does not always result in better generalization abilities. While providing agent capabilities information often improves the test-time performance on several tasks (as seen in Fig. 5), the corresponding generalization gap is worse in several instances. This indicates that the agents have overfitted to the training setting. The additional information has therefore assisted the task memorization rather than generalization. This phenomena is consistent with recent work which shows that recurrent networks such as LSTMs (used in the agent networks) are prone to memorization (Kirsch et al., 2022). Moreover, we hypothesise that grounding abilities remain a key challenge for current baselines, and better better grounding mechanisms in MARL algorithms (e.g., forward dynamics prediction as in Jaderberg et al. (2016) are required. The failure to generalize on index-based privileged information regarding agent types suggests using mechanisms such as latent embeddings to compose and reason about capabilities. Finally, a low test performance gap between agents having privileged information vs. those that do not, coupled with a low generalization gap, suggests that these methods do not facilitate information sharing between the agents, which is another desideratum towards attaining CG.

### 5.3 Making progress towards Combinatorial Generalization (CG)

The theoretical and empirical analysis above analysis motivates several directions to help solve the combinatorial generalization problem. From, the experiments, it is clear that the biggest challenge in attaining CG in

complex domains is that of making the agents understand how their capabilities affect the team returns, which we refer as "grounding". As we elucidate in Section 2 and Section 3, our work generalizes the successor feature framework to multi agent systems. One nice consequence of the analysis is that it informs the use of a latent space based approach to embed the capabilities and the observations, and model the interactions similar to Eq. (2),Eq. (3) in the latent space to learn better capability representations for attaining generalization. Further, as latent maps can be arbitrarily complex, this can also be used for learning in situations involving complex non-linear dependencies on capabilities. Finally, we can augment this latent representation learning process with more structural information about the capability context using approaches similar to (Gelada et al., 2019; Mahajan & Zhang, 2023) for the single agent RL scenario.

## 6 Related Work

**Multi-agent systems** (Claus & Boutilier, 1998; Busoniu et al., 2008) offer means to overcome theoretical barriers like exponential blow up in state-action space and compute resource requirements for large problems. MARL is a promising approach for training cooperative MAS. Recent progress in cooperative MARL (Lowe et al., 2017; Sunehag et al., 2017; Rashid et al., 2020; Mahajan et al., 2021) has demonstrated impressive applications in solving complex tasks in games such as StarCraft II (Samvelyan et al., 2019). Specialized methods which improve exploration in MARL have been proposed using hierarchical learning (Mahajan et al., 2019) and successor features (Gupta et al., 2021). Methods for factorizing the action space (Wang et al., 2020) have shown improvement in sample complexity. Iqbal et al. (2021) regularize value functions to share factors comprised of sub-groups of entities, in order to transfer knowledge across cooperative tasks. In the competitive/general sum MARL space (Lowe et al., 2017; OpenAI et al., 2019) have shown impressive performance on complex tasks. Vezhnevets et al. (2020) use an options framework to learn agents which generalize against different opponents. Czarnecki et al. (2020); Tuyls et al. (2020); Piliouras et al. (2021) explore the structural and theoretical properties of general payoff games. Mehta et al. (2023) provide domains for social generalization in MARL, similarly Ellis et al. (2022) provide scenarios for procedural generation in StarCraft. Samvelyan et al. (2023) uses an autocurriculum over procedurally generated environments and population of agents for training generelizable agents in two-player zero-sum settings.

**Ad-hoc coordination** was first formalised by Stone et al. (2010) by modelling the multi-agent problem as a single-agent task and using competency scores to measure agent compatibility. Methods for using explicit hard-coded protocols for adaptations were explored in Tambe (1997); Grosz & Kraus (1996). Opponent modelling for general games was explored in Stone et al. (2000); Markovitch & Reger (2005); Ledezma et al. (2004); He et al. (2016); Grover et al. (2018). Several approaches to the ad-hoc cooperation problem assume that the behaviour of other agents in the ensemble are fixed (Bowling & McCracken, 2005). Planning methods like Monte Carlo tree search are used for finding optimal adaptation policy from a fixed set of choices (Barrett et al., 2011; Albrecht et al., 2016; Albrecht & Stone, 2019). Nikolaidis et al. (2014) develop over this by enabling learning a set of behaviours for the adapting agent while performing the task with human agents instead of assuming that it is given beforehand. Recent methods allow a change in the behaviour of the other agents to ones picked from a fixed set and account for the possible non-stationarities using change point detection Hernandez-Leal et al. (2017); Ravula (2019). Gu et al. (2021) use information based regularizer to learn a single ad-hoc agent. However, these methods do not consider arbitrary learning for other agents. Furthermore, they do not focus on generalization to unseen agent capabilities.

**Generalization in RL** aims to develop approaches that generalize well to the novel, unseen scenarios after training (Kirk et al., 2022). Such methods avoid overfitting to seen tasks and can produce robust behaviour when deployed to novel settings. Recent work on generalization in single-agent RL make use of techniques such as data augmentation (Raileanu et al., 2021; Kostrikov et al., 2021), environment generation (Team et al., 2021; Parker-Holder et al., 2022; Samvelyan et al., 2021), encoding inductive biases (Higgins et al., 2017), and regularization (Cobbe et al., 2019). Tang et al. (2022) extend generalization across value function by conditioning on policy representations. Methods in multi-task RL (MTRL) (Borsa et al., 2016) focus on learning policies and representation for generalization across the single agent multi task setting. (Sodhani et al., 2021) use a meta data based context learning approach for generalization in MTRL. (Teh et al., 2017) use policy distillation for regularization of policy learning process in MTRL. Methods in contextual MDPs (Hallak et al., 2015; Zhang et al., 2020; Mahajan & Zhang, 2023) also provide generalization with guarantees.

Recent work also elucidate some of the fundamental bounds arising from computational complexity which prevents sample efficient generalization (Du et al., 2020; Ghosh et al., 2021; Malik et al., 2021).

## 7 Conclusion and Future work

In this work, we studied the generalization properties in multi-agent systems (MAS) following Markovian dynamics with a linear dependence of dynamics on the agent capabilities. We showed how the framework extends the successor feature setting to MAS. We explored performance bounds for various interesting scenarios arising in MAS including generalization, transfer, agent substitutions, approximate inference of capabilities and deviations in environment dynamics. Furthermore, we showed how the bounds can be extended to the Lipschitz reward setting and elucidated the most general form of rewards and how they make generalization difficult. Finally, we extensively tested the popular MARL algorithms on domains presenting a wide spectrum of hardness for CG. We saw that while some algorithms demonstrated preliminary CG on easier domains, all of the algorithms are insufficient towards ensuring CG on the challenging domains. We further highlighted how the first step towards ensuring CG should be ensuring proper grounding of agent capabilities. For future work, we aim to provide tighter bounds for CG for more general dynamics and create methods for better grounding of agent capabilities.

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

# A  Proofs

## A.1  Generalisation between team compositions

**Theorem 1** (Generalisation between team compositions). *Let team compositions $\mathcal{T}^x, \mathcal{T}^y \in \mathcal{C}^n$ with influence weights $a^x, a^y \in \Delta_{n-1}$, $s_{max} = \max_s \|W_R s\|_1$ , $V_{mid} = \frac{1}{2} \max_s V^*_{\mathcal{T}^y}(s)$, Then[5]:*

$$|V^*_{\mathcal{T}^x} - V^*_{\mathcal{T}^y}| \leq \frac{s_{max} + \gamma d V_{mid}}{\gamma(1-\gamma)} \Psi, \text{ where}$$

$$\Psi = \Big[| \sum_i a_i^x (\mathcal{T}_i^x - \mathcal{T}_i^y)|_\infty + | \sum_i (a_i^x - a_i^y)\mathcal{T}_i^y|_\infty \Big]$$

*Proof.* Let $\epsilon_R = \max_s |R_{\mathcal{T}^x}(s) - R_{\mathcal{T}^y}(s)|$ and $\epsilon_P = \max_{s,\mathbf{u}} 2 \cdot D_{TV}\Big(P_{\mathcal{T}^x}(\cdot|s,\mathbf{u}), P_{\mathcal{T}^y}(\cdot|s,\mathbf{u})\Big)$ where $D_{TV}$ is the total variation distance. We have that:

$$|Q^*_{\mathcal{T}^x}(s,\mathbf{u}) - Q^*_{\mathcal{T}^y}(s,\mathbf{u})|$$

$$= |R_{\mathcal{T}^x}(s) - R_{\mathcal{T}^y}(s) + \gamma \Big( \sum_{s'} P_{\mathcal{T}^x}(s'|s,\mathbf{u}) \max_{u'} Q^*_{\mathcal{T}^x}(s',\mathbf{u}') - \sum_{s'} P_{\mathcal{T}^y}(s'|s,\mathbf{u}) \max_{u'} Q^*_{\mathcal{T}^y}(s',\mathbf{u}')) \Big)|$$

$$\leq |R_{\mathcal{T}^x}(s) - R_{\mathcal{T}^y}(s)| + \gamma \Big\{ | \sum_{s'} P_{\mathcal{T}^x}(s'|s,\mathbf{u}) \Big[ \max_{u'} Q^*_{\mathcal{T}^x}(s',\mathbf{u}') - \max_{u'} Q^*_{\mathcal{T}^y}(s',\mathbf{u}') \Big]|$$

$$+ | \sum_{s'} \Big[ P_{\mathcal{T}^x}(s'|s,\mathbf{u}) - P_{\mathcal{T}^y}(s'|s,\mathbf{u}) \Big] (\max_{u'} Q^*_{\mathcal{T}^y}(s',\mathbf{u}') - V_{mid})| \Big\}$$

$$\leq \epsilon_R + \gamma \Big\{ \sum_{s'} P_{\mathcal{T}^x}(s'|s,\mathbf{u}) | \max_{u'} Q^*_{\mathcal{T}^x}(s',\mathbf{u}') - \max_{u'} Q^*_{\mathcal{T}^y}(s',\mathbf{u}')| + \sum_{s'} |P_{\mathcal{T}^x}(s'|s,\mathbf{u}) - P_{\mathcal{T}^y}(s'|s,\mathbf{u})|| \max_{u'} Q^*_{\mathcal{T}^y}(s',\mathbf{u}') - V_{mid}| \Big\}$$

$$\leq \epsilon_R + \gamma \Big\{ \sum_{s'} P_{\mathcal{T}^x}(s'|s,\mathbf{u}) \max_{u'} |Q^*_{\mathcal{T}^x}(s',\mathbf{u}') - Q^*_{\mathcal{T}^y}(s',\mathbf{u}')| + 2 \cdot D_{TV}\Big(P_{\mathcal{T}^x}(s'|s,\mathbf{u}), P_{\mathcal{T}^y}(s'|s,\mathbf{u})\Big) V_{mid} \Big\}$$

$$\leq \epsilon_R + \gamma \Big\{ \max_{s',u'} |Q^*_{\mathcal{T}^x}(s',\mathbf{u}') - Q^*_{\mathcal{T}^y}(s',\mathbf{u}')| + \epsilon_P V_{mid} \Big\}$$

Next taking max w.r.t. $s, u$ of the above we get:

$$\max_{s,u} |Q^*_{\mathcal{T}^x}(s,\mathbf{u}) - Q^*_{\mathcal{T}^y}(s,\mathbf{u})| \leq \frac{\epsilon_R + \gamma \epsilon_P V_{mid}}{1-\gamma}$$

We now bound the deviation quantities appearing above:

$$\epsilon_R = \max_s |R_{\mathcal{T}^x}(s) - R_{\mathcal{T}^y}(s)|$$

$$= \max_s | \sum_{i=1}^n a_i^x \langle \mathcal{T}_i^x \cdot W_R s \rangle - \sum_{i=1}^n a_i^y \langle \mathcal{T}_i^y \cdot W_R s \rangle |$$

$$\leq \max_s \Big[ | \sum_{i=1}^n a_i^x \langle (\mathcal{T}_i^x - \mathcal{T}_i^y) \cdot W_R s \rangle | + | \sum_{i=1}^n (a_i^x - a_i^y) \langle \mathcal{T}_i^y \cdot W_R s \rangle | \Big]$$

$$\leq \max_s \Big[ | \sum_i a_i^x (\mathcal{T}_i^x - \mathcal{T}_i^y)|_\infty |W_R s|_1 + | \sum_i (a_i^x - a_i^y)\mathcal{T}_i^y|_\infty |W_R s|_1 \Big]$$

$$= s_{max} \Big[ | \sum_i a_i^x (\mathcal{T}_i^x - \mathcal{T}_i^y)|_\infty + | \sum_i (a_i^x - a_i^y)\mathcal{T}_i^y|_\infty \Big]$$

Similarly,

$$\epsilon_P = \max_{s,\mathbf{u}} 2 \cdot D_{TV}\Big(P_{\mathcal{T}^x}(\cdot|s,\mathbf{u}), P_{\mathcal{T}^y}(\cdot|s,\mathbf{u})\Big)$$

---

[5]for $\gamma \in (0, \frac{\sqrt{5}-1}{2})$ we can replace $\frac{1}{\gamma(1-\gamma)}$ by $\frac{1+\gamma}{1-\gamma}$

$$= \max_{s,\mathbf{u}} \sum_{s'} |P_{\mathcal{T}^x}(s'|s,\mathbf{u}) - P_{\mathcal{T}^y}(s'|s,\mathbf{u})|$$

$$= \max_{s,\mathbf{u}} \sum_{s'} |\sum_{i=1}^{n} a_i^x \langle \mathcal{T}_i^x \cdot W_P(s',s,\mathbf{u})\rangle - \sum_{i=1}^{n} a_i^y \langle \mathcal{T}_i^y \cdot W_P(s',s,\mathbf{u})\rangle|$$

$$\leq \max_{s,\mathbf{u}} \sum_{s'} \Big[|\sum_{i=1}^{n} a_i^x \langle(\mathcal{T}_i^x - \mathcal{T}_i^y) \cdot W_P(s',s,\mathbf{u})\rangle| + |\sum_{i=1}^{n} (a_i^x - a_i^y)\langle \mathcal{T}_i^y \cdot W_P(s',s,\mathbf{u})\rangle|\Big]$$

$$\leq \max_{s,\mathbf{u}} \sum_{s'} \Big[|\sum_{i} a_i^x (\mathcal{T}_i^x - \mathcal{T}_i^y)|_\infty |W_P(s',s,\mathbf{u})|_1 + |\sum_{i} (a_i^x - a_i^y)\mathcal{T}_i^y|_\infty |W_P(s',s,\mathbf{u})|_1\Big]$$

$$= \Big[|\sum_{i} a_i^x (\mathcal{T}_i^x - \mathcal{T}_i^y)|_\infty + |\sum_{i} (a_i^x - a_i^y)\mathcal{T}_i^y|_\infty\Big] \max_{s,\mathbf{u}} \sum_{s'} |W_P(s',s,\mathbf{u})|_1$$

$$= d\Big[|\sum_{i} a_i^x (\mathcal{T}_i^x - \mathcal{T}_i^y)|_\infty + |\sum_{i} (a_i^x - a_i^y)\mathcal{T}_i^y|_\infty\Big]$$

Thus, we get:

$$|Q_{\mathcal{T}^x}^*(s,\mathbf{u}) - Q_{\mathcal{T}^y}^*(s,\mathbf{u})| \leq \frac{s_{max} + \gamma dV_{mid}}{1-\gamma}\Big[|\sum_{i} a_i^x (\mathcal{T}_i^x - \mathcal{T}_i^y)|_\infty + |\sum_{i} (a_i^x - a_i^y)\mathcal{T}_i^y|_\infty\Big]$$

Finally we get the value difference bound by considering a dummy state $s^{\#}$ which always transitions according to $\rho$ and then using the Bellman equation. (Note that for $\gamma \in (0, \frac{\sqrt{5}-1}{2})$ we can replace $\frac{1}{\gamma(1-\gamma)}$ by $\frac{1+\gamma}{1-\gamma}$ for a tighter bound without considering a dummy start state) □

**Corollary 1.1** (Change in optimal value as a result of agent substitution)**.** *Let $\mathcal{T} \in \mathcal{C}^n$ be a team composition with influence weights $a \in \Delta_{n-1}$. If agent $i$ is substituted with $i'$ keeping $a_i$ unchanged such that $|\mathcal{T}_{i'} - \mathcal{T}_i|_\infty \leq \epsilon_C$ then the new team $(\mathcal{T}')$ optimal value follows:*

$$|V_{\mathcal{T}'}^* - V_{\mathcal{T}}^*| \leq \frac{(s_{max} + \gamma dV_{mid})a_i\epsilon_C}{\gamma(1-\gamma)}$$

*Proof.* Applying Theorem 1 on original task and a new task with same influence weights and agent $i$ capability replaced with $\mathcal{T}_{i'}$ immediately gives the result. □

## A.2 Transfer of optimal policy

**Theorem 2** (Transfer of optimal policy)**.** *Let $\mathcal{T}^x, \mathcal{T}^y \in \mathcal{C}^n$, $a^x, a^y \in \Delta_{n-1}$, $s_{max} = \max_s ||W_R s||_1$, $V_{mid} = \frac{1}{2}\max_s V_{\mathcal{T}^y}^*(s)$. Let $\pi_y^*$ be the optimal policy for the team composed of agents with capabilities $\mathcal{T}^y$ and influence weights $a^y$. Then:*

$$V_{\mathcal{T}^x}^* - V_{\mathcal{T}^x}^{\pi_y^*} \leq 2\frac{s_{max} + \gamma dV_{mid}}{\gamma(1-\gamma)}\Psi,$$

*where $\Psi$ is defined as in Eq. (4).*

*Proof.* We have that:

$$Q_{\mathcal{T}^x}^*(s,\mathbf{u}) - Q_{\mathcal{T}^x}^{\pi_y^*}(s,\mathbf{u}) \leq |Q_{\mathcal{T}^x}^*(s,\mathbf{u}) - Q_{\mathcal{T}^y}^*(s,\mathbf{u})| + |Q_{\mathcal{T}^y}^*(s,\mathbf{u}) - Q_{\mathcal{T}^x}^{\pi_y^*}(s,\mathbf{u})| \tag{6}$$

The first term on the RHS of Eq. (6) is taken care of by Theorem 1. We now focus on the second term:

$$|Q_{\mathcal{T}^y}^*(s,\mathbf{u}) - Q_{\mathcal{T}^x}^{\pi_y^*}(s,\mathbf{u})|$$

$$= |R_{\mathcal{T}^y}(s) - R_{\mathcal{T}^x}(s) + \gamma\Big(\sum_{s'} P_{\mathcal{T}^y}(s'|s,\mathbf{u})\max_{u'} Q_{\mathcal{T}^y}^*(s',\mathbf{u}') - \sum_{s'} P_{\mathcal{T}^x}(s'|s,\mathbf{u})Q_{\mathcal{T}^x}^{\pi_y^*}(s',\pi_y^*(\mathbf{u}'))\Big)|$$

$$\leq \epsilon_R + \gamma\Big\{|\sum_{s'} P_{\mathcal{T}^x}(s'|s,\mathbf{u})\Big[\max_{u'} Q_{\mathcal{T}^y}^*(s',\mathbf{u}') - Q_{\mathcal{T}^x}^{\pi_y^*}(s',\pi_y^*(\mathbf{u}'))\Big]|$$

$$+ |\sum_{s'} \Big[ P_{\mathcal{T}^y}(s'|s,\mathbf{u}) - P_{\mathcal{T}^x}(s'|s,\mathbf{u}) \Big] (\max_{u'} Q^*_{\mathcal{T}^y}(s',\mathbf{u}') - V_{mid}) | \Big\}$$

$$\leq \epsilon_R + \gamma \Big\{ \max_{s',u'} |Q^*_{\mathcal{T}^y}(s',\mathbf{u}') - Q^{\pi^*_y}_{\mathcal{T}^x}(s',\pi^*_y(\mathbf{u}')| + \epsilon_P V_{mid} \Big\}$$

Once again, taking max w.r.t. $s, \mathbf{u}$ of the above we get:

$$\max_{s,u} |Q^*_{\mathcal{T}^y}(s,\mathbf{u}) - Q^{\pi^*_y}_{\mathcal{T}^x}(s,\mathbf{u})| \leq \frac{\epsilon_R + \gamma \epsilon_P V_{mid}}{1 - \gamma}$$

Substituting for deviation expressions and using Theorem 1 in Eq. (6) we get:

$$|Q^*_{\mathcal{T}^x}(s,\mathbf{u}) - Q^{\pi^*_y}_{\mathcal{T}^x}(s,\mathbf{u})| \leq 2 \frac{s_{max} + \gamma dV_{mid}}{1 - \gamma} \Big[ |\sum_i a^x_i (\mathcal{T}^x_i - \mathcal{T}^y_i)|_\infty + |\sum_i (a^x_i - a^y_i)\mathcal{T}^y_i|_\infty \Big]$$

Note the absolute on LHS above can be dropped as $Q^*_{\mathcal{T}^x}$ is optimal. Finally using the same technique as above for Theorem 1 we get the statement of the theorem. □

**Corollary 2.1** (Out of distribution performance). *Let $\mathcal{T} \notin Sup(\mathcal{M})$ be an out of distribution task, we then have that the performance of the absolute oracle policy on $\mathcal{T}$ satisfies:*

$$V^*_{\mathcal{T}} - V^{\pi^*_{\mathcal{M}}}_{\mathcal{T}} \leq 2 \frac{s_{max} + \gamma dV_{mid}}{\gamma(1 - \gamma)} d_a(\mathcal{T}, Sup(\mathcal{M})),$$

*Proof.* For any task that belongs to $\arg\min_{\mathcal{T}^l \in Sup(\mathcal{M})} d_a(\mathcal{T}^l, \mathcal{T})$, we have by application of Theorem 2 that the result immediately holds given definition of $\pi^*_{\mathcal{M}}$. □

## A.3 Population decrease

**Theorem 3** (Population decrease bound). *For the team composition $\mathcal{T} \in \mathcal{C}^n$ with influence weights $a \in \Delta_{n-1}$. If agent $n$ is eliminated followed by a re-normalization of influence weights, we have that for the remaining team ($\mathcal{T}^- \triangleq (\mathcal{T})_{i=1}^{n-1}$):*

$$|V^*_{\mathcal{T}^-} - V^*_{\mathcal{T}}| \leq \frac{a_n(s_{max} + \gamma dV_{mid})}{\gamma(1 - \gamma)} \Big| \sum_{i=1}^{n-1} \frac{a_i \mathcal{T}_i}{1 - a_n} - \mathcal{T}_n \Big|_\infty$$

*Proof.* We use Theorem 1 with influence weights $(a_i)_1^n$ and $(\lambda \cdot a_i : i = 1..n-1, a_n = 0)$ where $\lambda = \frac{1}{1-a_n}$ □

**Corollary 3.1** (Population increase bound). *For the team composition $\mathcal{T} \in \mathcal{C}^n$ with influence weights $a \in \Delta_{n-1}$. If agent $n+1$ is added with capability $\mathcal{T}_{n+1}$ and weight $a_{n+1}$ (other weights scaled down by $\lambda = 1 - a_{n+1}$) we have that for the new team ($\mathcal{T}^+ \triangleq (\mathcal{T}_1..\mathcal{T}_n, \mathcal{T}_{n+1})$):*

$$|V^*_{\mathcal{T}^+} - V^*_{\mathcal{T}}| \leq \frac{a_{n+1}(s_{max} + \gamma dV_{mid})}{\gamma(1 - \gamma)} \Big| \sum_{i=1}^{n} a_i \mathcal{T}_i - \mathcal{T}_{n+1} \Big|_\infty$$

*Proof.* Consider the team compositions $\mathcal{T}^x = (\mathcal{T}_1..\mathcal{T}_n, 0)$ with influence weights $= (a_1..a_n, 0)$ and $\mathcal{T}^y = (\mathcal{T}_1..\mathcal{T}_n, \mathcal{T}_{n+1})$ with influence weights $= (\lambda a_1..\lambda a_n, a_{n+1})$ where $\lambda = 1 - a_{n+1}$, we have that:

$$\Psi = \Big[ |\sum_i a^x_i (\mathcal{T}^x_i - \mathcal{T}^y_i)|_\infty + |\sum_i (a^x_i - a^y_i)\mathcal{T}^y_i|_\infty \Big]$$

$$= |\sum_{i=1}^{n} (1 - \lambda)a_i \mathcal{T}^y_i - a_{n+1}\mathcal{T}^y_{n+1}|_\infty$$

$$= a_{n+1}|\sum_{i=1}^{n} a_i \mathcal{T}^y_i - \mathcal{T}^y_{n+1}|_\infty$$

which on applying Theorem 1 yields the result. □

### A.4 Approximate $\hat{\epsilon}_R, \hat{\epsilon}_P$ dynamics

**Theorem 4** (Approximate $\hat{\epsilon}_R, \hat{\epsilon}_P$ dynamics). *Let $\mathcal{T}^x, \mathcal{T}^y \in \mathcal{C}^n$, $a^x, a^y \in \Delta_{n-1}$ and the dynamics be only approximately linear so that $|R_\mathcal{T}(s) - \sum_{i=1}^n a_i \langle c_i \cdot W_R s \rangle| \leq \hat{\epsilon}_R$ and $|P_\mathcal{T}(s'|s, \mathbf{u}) - \sum_{i=1}^n a_i \langle c_i \cdot W_P(s', s, \mathbf{u}) \rangle| \leq \hat{\epsilon}_P$. Then:*

$$|V_{\mathcal{T}^x}^* - V_{\mathcal{T}^y}^*| \leq \frac{s_{max} + \gamma d V_{mid}}{\gamma(1-\gamma)} \Psi + \frac{2(\hat{\epsilon}_R + \gamma \hat{\epsilon}_P V_{mid})}{\gamma(1-\gamma)},$$

*where $\Psi$ is defined as in Eq. (4).*

*Proof.* We begin as in proof of Theorem 1 to get:

$$\max_{s,u} |Q_{\mathcal{T}^x}^*(s, \mathbf{u}) - Q_{\mathcal{T}^y}^*(s, \mathbf{u})| \leq \frac{\epsilon_R + \gamma \epsilon_P V_{mid}}{1-\gamma}$$

Next we apply the corrections to the relative differences:

$$\epsilon_R = \max_s |R_{\mathcal{T}^x}(s) - R_{\mathcal{T}^y}(s)|$$

$$\leq \max_s \left[ |R_{\mathcal{T}^x}(s) - \sum_{i=1}^n a_i^x \langle \mathcal{T}_i^x \cdot W_R s \rangle| + |\sum_{i=1}^n a_i^x \langle \mathcal{T}_i^x \cdot W_R s \rangle - \sum_{i=1}^n a_i^y \langle \mathcal{T}_i^y \cdot W_R s \rangle| + |R_{\mathcal{T}^y}(s) - \sum_{i=1}^n a_i^y \langle \mathcal{T}_i^y \cdot W_R s \rangle| \right]$$

$$\leq 2\hat{\epsilon}_R + \max_s \left[ |\sum_{i=1}^n a_i^x \langle (\mathcal{T}_i^x - \mathcal{T}_i^y) \cdot W_R s \rangle| + |\sum_{i=1}^n (a_i^x - a_i^y) \langle \mathcal{T}_i^y \cdot W_R s \rangle| \right]$$

$$\leq 2\hat{\epsilon}_R + \max_s \left[ |\sum_i a_i^x (\mathcal{T}_i^x - \mathcal{T}_i^y)|_\infty |W_R s|_1 + |\sum_i (a_i^x - a_i^y) \mathcal{T}_i^y|_\infty |W_R s|_1 \right]$$

$$= 2\hat{\epsilon}_R + s_{max} \left[ |\sum_i a_i^x (\mathcal{T}_i^x - \mathcal{T}_i^y)|_\infty + |\sum_i (a_i^x - a_i^y) \mathcal{T}_i^y|_\infty \right]$$

Proceeding similarly with the transition probabilities we get the desired result. $\qquad \square$

### A.5 Error from estimation of capabilities

**Theorem 5** (Error from estimation of capabilities). *For the team composition $\mathcal{T} \in \mathcal{C}^n$ with influence weights $a \in \Delta_{n-1}$. If the agent capabilities are inaccurately inferred as $\hat{\mathcal{T}}$ with $\max_i |\mathcal{T}_i - \hat{\mathcal{T}}_i|_\infty \leq \epsilon_\mathcal{T}$ and agents learn the inexact policy $\hat{\pi}^*$ then:*

$$|V_\mathcal{T}^* - V_\mathcal{T}^{\hat{\pi}^*}| \leq \frac{2\epsilon_\mathcal{T}(s_{max} + \gamma d V_{mid})}{\gamma(1-\gamma)}$$

*where $V_{mid} = \frac{1}{2} \max_s V_{\hat{\mathcal{T}}}^*(s)$*

*Proof.* We have that for the actual and inferred team compositions with same influence weights:

$$\Psi = \left[ |\sum_i a_i (\mathcal{T}_i - \hat{\mathcal{T}}_i)|_\infty + |\sum_i (a_i - a_i) \hat{\mathcal{T}}_i|_\infty \right]$$

$$= |\sum_i a_i (\mathcal{T}_i - \hat{\mathcal{T}}_i)|_\infty$$

$$\leq \sum_i a_i |\mathcal{T}_i - \hat{\mathcal{T}}_i|_\infty$$

$$\leq \sum_i a_i \epsilon_\mathcal{T}$$

$$= \epsilon_\mathcal{T}$$

Now applying Theorem 2 gives the result $\qquad \square$

## A.6 Extending to Lipschitz rewards

We demonstrate how to extend the results in Section 3 to Lipschitz function of capabilities. For brevity we consider only the setting where the rewards vary with capabilities. Thus, for the reward function form $R_{\mathcal{T}}(s) = \langle f(\mathcal{T}) \cdot W_R s \rangle$ where $f(\mathcal{T})$ is $L_i$ Lipschitz with respect to the capability $\mathcal{T}_i$ for $i \in \mathcal{A}$ for the $|\cdot|_\infty$ norm. We get that for two different team compositions $\mathcal{T}^x, \mathcal{T}^y$

$$
\begin{aligned}
\epsilon_R &= \max_s |R_{\mathcal{T}^x}(s) - R_{\mathcal{T}^y}(s)| \\
&= \max_s |\langle f(\mathcal{T}^x) \cdot W_R s \rangle - \langle f(\mathcal{T}^y) \cdot W_R s \rangle| \\
&= \max_s |\sum_{i=1}^n \langle f(\mathcal{T}^i) \cdot W_R s \rangle - \langle f(\mathcal{T}^{i+1}) \cdot W_R s \rangle| \\
&\leq \max_s \sum_{i=1}^n |\langle f(\mathcal{T}^i) \cdot W_R s \rangle - \langle f(\mathcal{T}^{i+1}) \cdot W_R s \rangle| \\
&\leq \max_s \sum_{i=1}^n |\langle f(\mathcal{T}^i) \cdot W_R s \rangle - \langle f(\mathcal{T}^{i+1}) \cdot W_R s \rangle| \\
&\leq \max_s \sum_{i=1}^n |f(\mathcal{T}^i) - f(\mathcal{T}^{i+1})|_\infty |W_R s|_1 \\
&\leq s_{max} \sum_{i=1}^n L_i |\mathcal{T}_i^x - \mathcal{T}_i^y|_\infty
\end{aligned}
$$

Where $\mathcal{T}^i$ was the sequence satisfying $\mathcal{T}^1 = \mathcal{T}^x$ and $\mathcal{T}^{n+1} = \mathcal{T}^y$ and changing $\mathcal{T}^x$ one index at a time. We have thus proved that:

**Theorem 6.** *For rewards $L_i$ Lipschitz in the capabilities with respect to $|\cdot|_\infty$ norm, the difference in optimal values between team compositions $\mathcal{T}^x, \mathcal{T}^y$ satisfy:*

$$
|V_{\mathcal{T}^x}^* - V_{\mathcal{T}^y}^*| \leq \frac{s_{max} \sum_{i=1}^n L_i |\mathcal{T}_i^x - \mathcal{T}_i^y|_\infty}{\gamma(1-\gamma)}
$$

## A.7 General dependence of rewards on capabilities:

We now consider the dependence of rewards on the capabilities in the most general form. For this, we introduce the notion of $(\alpha, k)$-rewards where $\alpha \geq 0, k \in \mathbb{N}$.

$$
R_{\mathcal{T}}(s) = \left\langle \sum_{k_i \in \mathbb{N}, \sum k_i \leq k} a_{k_1..k_n} \Pi_{i=1}^n c_i^{k_i} \cdot W_R s \right\rangle \tag{7}
$$

where $\mathbb{N}$ are non negative integers, $|a_{k_1..k_n}| \leq \alpha$ and $c_i^{k_i}$ represents element-wise exponentiation. . Rewards in Eq. (2) can be seen as a special case belonging to Eq. (7) the choice $\alpha, k = 1$. Similarly the union $\cup_{\alpha \geq 0, k \in \mathbb{N}}(\alpha, k)$-rewards cover all possible reward dependencies on capabilities. We have further relaxed the assumption of influence weights belonging to a simplex here and replaced it with individual bounds on the power series coefficients here. We next see that for this scenario, even a small change in the capability of a single agent can shift the rewards massively. Let the capability of agent $i$ be changed from $\mathcal{T}_i$ to $\mathcal{T}_{i'}$ such that $|\mathcal{T}_i - \mathcal{T}_{i'}|_\infty \leq \delta$. Then we have

**Lemma 1.** *For substitution $\mathcal{T}_i$ to $\mathcal{T}_{i'}$ such that $|\mathcal{T}_i - \mathcal{T}_{i'}|_\infty \leq \delta$ under the $(\alpha, k)$-rewards setting we have that*

$$
\begin{aligned}
\epsilon_R &= \max_{s \in S} \left| \langle f(\mathcal{T}^x) \cdot W_R s \rangle - \langle f(\mathcal{T}^y) \cdot W_R s \rangle \right| \\
&= \max_{s \in S} \left| \left\langle \sum_{k_i \in \mathbb{N}, \sum k_i \leq k} a_{k_1..k_n} \Pi_{j \neq i} \mathcal{T}_j^{k_j} (\mathcal{T}_i^{k_i} - \mathcal{T}_{i'}^{k_i}) \cdot W_R s \right\rangle \right|
\end{aligned}
$$

$$\leq \max_{s \in S} \left| \sum_{k_i \in \mathbb{N}, \sum k_i \leq k} a_{k_1..k_n} \Pi_{j \neq i} \mathcal{T}_j^{k_j} (\mathcal{T}_i^{k_i} - \mathcal{T}_{i'}^{k_i}) \right|_\infty \left| W_R s \right|_1$$

$$\leq \alpha s_{max} \sum_{j=0}^{k} \sum_{l=1}^{j} \binom{l}{j} l |\mathcal{T}_i^{k_i} - \mathcal{T}_{i'}^{k_i}|_\infty$$

$$\leq \alpha \delta s_{max} \sum_{j=0}^{k} j 2^{j-1} = \mathcal{O}(\alpha \delta s_{max} k 2^k)$$

*The above gives us:*

$$|V_{\mathcal{T}^x}^* - V_{\mathcal{T}^y}^*| \leq \frac{\mathcal{O}(\alpha \delta s_{max} k 2^k)}{\gamma(1-\gamma)}$$

*where $\mathcal{T}^x, \mathcal{T}^y$ are the joint capabilities before and after agent i capability is changed respectively and $\mathcal{O}(\cdot)$ denotes the order of the term.*

While this is not a lower bound, the above still suggests that even a small change in the capability of an agent can cause the rewards to change by a lot, hence it is natural to expect that generalization becomes harder as the problem start showing the needle in the haystack phenomenon where only the *right combination* of capabilities gives a large optimal value.

## B  Experimental Setup

### B.1  Environments

### B.1.1  Fruit Forage

We use the fruit forage task on a grid world to empirically demonstrate the generalisation bounds in Section 3. On a $k \times k$ grid world we have $n$ agents and $d$ types of fruit trees. For each agent $i$, $\mathcal{T}_i(j), j \in \{1..d\}$ represents the utility of fruit $j$ for agent $i$. The state vector is appended with the $d$ dimensional binary vector representing whether each of the tree types was foraged at a given time step. The details for the team compositions can be found in Appendix B.1.1. We define three team compositions as follows:

$T_x$: [[0.05, 0.1, 0.6, 2.8], [0.05, 0.1, 2.1, 0.8], [0.05, 0.1, 1.8, 1.2], [0.05, 0.1, 0.9, 2.4]]

$T_y$: [[0.7, 0.4, 0.15, 0.2], [0.2, 1.4, 0.15, 0.2], [0.3, 1.2, 0.15, 0.2], [0.6, 0.6, 0.15, 0.2]]

$T_z$: [[0.1, 0.3, 0.6, 0.0], [0.4, 0.1, 0.5, 0.0], [0.05, 0.06, 0.89, 0.0], [0.0, 0.0, 0.0, 1.0]]

For proving bounds on Theorem-1, we compare the mean test returns achieved on tasks $T_x$ and $T_y$ using $V_{T_x}^\star - V_{T_y}^\star$. For Theorem-2, we compare the mean test returns achieved on tasks $T_x$ and optimal policies of task $T_y$ evaluated on task $T_x$ i.e. $V_{T_x}^\star - V_{T_x}^{\pi_{T_y}^\star}$. Finally, for Theorem-3, we compare the mean test returns achieved on tasks $T_z$ and optimal policies of task $T_z$ evaluated on task $T_z$ but removing the last agent i.e. $V_{T_{z-}}^\star - V_{T_z}^\star$.

### B.1.2  Predator Prey

We consider a complicated partially observable predator-prey (PP) task in an $8 \times 8$ grid involving four agents (predators) and four prey that is designed to test coordination between agents. Specifically, each predator has a parameter describing the hit point damage it can cause the prey. Similarly, the prey comes with variations in health. For example, a prey with a capability of 5 can only be caught if the total capability of agents taking the capture action simultaneously on it have capabilities $\geq 5$ (such as [1,1,3]), otherwise, the whole team receives a penalty $p$. On successful capture, agents get a reward of +1. Once prey is captured, another prey is spawned at a random location. Therefore, agents have to collaborate and capture as many preys as possible within 100 time steps.

Each agent can take 6 actions i.e. move in one of the 4 directions (Up, Left, Down, Right), remain still (no-op), or try to catch (capture) any adjacent prey. The prey moves around in the grid with a probability of 0.7 and remains still at its position with the probability of 0.3. Impossible actions for both agents and prey are marked unavailable, for eg. moving into an occupied cell or trying to take a capture action with no adjacent prey.

In this domain, we test for two types of generalization: (1) novel team composition where test tasks contain a team composition which has not been encountered during training (PP Unseen Team in Figure 4), and second, (2) test tasks where novel team compositions can also have agent types with capabilities not encountered during training (PP Unseen Team, Agent in Figure 4).

For (PP Unseen Team), we train on preys with capabilities [2,2,2,3], and agents with capabilities [2,3,2,3],[1,2,1,2], thereby having agent teams with total hit points of 10 and 6 respectively. We also train on two separate penalties $p$ for miscoordination i.e. $p \in \{0.0, -0.008\}$, this helps inject additional stochasticity in the environment as the agents don't know the penalty value. For test tasks, we create novel team compositions not encountered during training i.e. agents with capabilities [1,1,2,3],[1,1,1,3] having total hit points of 7 and 6 respectively.

For (PP Unseen Team, Agent) we train on preys with capabilities [1,2,3,4], and agents with capabilities [1, 2, 2, 3], [1, 1, 2, 2], [1, 3, 2, 1], thereby having agent teams with total hit points of 8, 6 and 7 respectively. We also train on two separate penalties $p$ for miscoordination i.e. $p \in \{0.0, -0.008\}$. For test tasks, we create novel team compositions with an unseen agent of capability 4 not encountered during training i.e. agents with capabilities [1, 1, 1, 4], [1, 1, 3, 4], [1, 1, 2, 4] having total hit points of 7, 9, and 8 respectively.

**Experimental Setup:** For (PP Unseen Team, and PP Unseen Team, Agent) oracle baseline (leftmost), we show the average difference in performance across all test tasks when capability information is included ((c) for each method.

For testing the generalization gap in (PP Unseen Team), we show the difference in returns achieved by training task [1,2,1,2] (hit point 6) and test task [1,1,1,3] (hit point 6). For testing the generalization gap in (PP Unseen Team, Agent), we show the difference in returns achieved by training task [1,3,2,1] (hit point 7) and test task [1,1,1,4] (hit point 7) with a new agent of capability 4. All PP experiments are based on 8 seeds.

### B.1.3  StarCraft II

We use the standard set of actions and global state information included as part of the SMAC benchmark Samvelyan et al. (2019). The sight range of the agent units has been increased to the fully observable setting. In the oracle mode, agent capabilities are included as part of individual observations. Each agent always observes its own capabilities. Furthermore, capabilities are always included in the global state.

`10_Terran` and `10_Terran_Hard` environment includes Marine, Maradeur, and Medivac units. `10_Protoss` and `10_Protoss_Hard` environments feature Stalker, Zealot, and Colossus units. `10_Zerg` and `10_Zerg_Hard` environments include Zergling, Hydralisk and Baneling units.

In `Accuracy` and `Health` tasks, specific values reduced from full unit capabilities are chosen to be equivalent to a loss of a single teammate. For example, if there three agents, their accuracy could be set to 0.75, 0.75 and 0.5 given that $(1 - 0.5) + (1 - 0.75) + (1 - 0.75) = 1$. Consequently, the overall reduction in accuracy would be roughly equivalent to losing one ally unit. This was chosen to ensure that the difficulty of the tasks was not too high.

All SMAC experiments are based on 5 seeds.

Table 1, 2, and 3 describe the training and evaluation distributions used in unit type swapping tasks.

Table 1: Team formations in `Terran` tasks

| 10_Terran | 10_Terran_Hard |
|---|---|
| **Training** | **Training** |
| 1 marine & 9 marauders | 1 marine & 9 marauders |
| 3 marines & 7 marauders | 2 marines & 8 marauders |
| 4 marines & 6 marauders | 3 marines & 7 marauders |
| 5 marines & 5 marauders | 4 marines & 6 marauders |
| 6 marines & 4 marauders | 5 marines & 5 marauders |
| 8 marines & 2 marauders | 6 marines & 4 marauders |
| 9 marines & 1 marauder | 7 marines & 3 marauders |
| 5 marauders & 5 medivacs | 8 marines & 2 marauders |
| 7 marauders & 3 medivacs | 9 marines & 1 marauder |
| 9 marauders & 1 medivac | 5 marauders & 5 medivacs |
| 7 marines & 3 medivacs | 6 marauders & 4 medivacs |
| 8 marines & 2 medivacs | 7 marauders & 3 medivacs |
| 9 marines & 1 medivac | 8 marauders & 2 medivacs |
| 10 marines | 9 marauders & 1 medivac |
| 10 marauders | 7 marines & 3 medivacs |
| 8 marines & 1 marauder & 1 medivac | 8 marines & 2 medivacs |
| 1 marine & 8 marauders & 1 medivac | 9 marines & 1 medivac |
| 5 marines & 3 marauders & 2 medivacs | **Testing** |
| 2 marines & 7 marauders & 1 medivac | 10 marines |
| 6 marines & 2 marauders & 2 medivacs | 10 marauders |
| 2 marines & 6 marauders & 2 medivacs | 8 marines & 1 marauder & 1 medivac |
| 4 marines & 4 marauders & 2 medivacs | 1 marine & 8 marauders & 1 medivac |
| **Testing** | 5 marines & 3 marauders & 2 medivacs |
| 2 marines & 8 marauders | 3 marines & 5 marauders & 2 medivacs |
| 7 marines & 3 marauders | 4 marines & 3 marauders & 3 medivacs |
| 6 marauders & 4 medivacs | 3 marines & 4 marauders & 3 medivacs |
| 8 marauders & 2 medivacs | 7 marines & 2 marauders & 1 medivac |
| 3 marines & 5 marauders & 2 medivacs | 2 marines & 7 marauders & 1 medivac |
| 4 marines & 3 marauders & 3 medivacs | 6 marines & 2 marauders & 2 medivacs |
| 3 marines & 4 marauders & 3 medivacs | 2 marines & 6 marauders & 2 medivacs |
| 7 marines & 2 marauders & 1 medivac | 4 marines & 4 marauders & 2 medivacs |

Table 2: Team formations in `Zerg` tasks

| 10_Zerg | 10_Zerg_Hard |
|---|---|
| **Training** | **Training** |
| 1 zergling & 9 hydralisks | 1 zergling & 9 hydralisks |
| 2 zerglings & 8 hydralisks | 2 zerglings & 8 hydralisks |
| 4 zerglings & 6 hydralisks | 3 zerglings & 7 hydralisks |
| 5 zerglings & 5 hydralisks | 4 zerglings & 6 hydralisks |
| 6 zerglings & 4 hydralisks | 5 zerglings & 5 hydralisks |
| 7 zerglings & 3 hydralisks | 6 zerglings & 4 hydralisks |
| 9 zerglings & 1 hydralisk | 7 zerglings & 3 hydralisks |
| 4 hydralisks & 6 banelings | 8 zerglings & 2 hydralisks |
| 5 hydralisks & 5 banelings | 9 zerglings & 1 hydralisk |
| 6 hydralisks & 4 banelings | 4 hydralisks & 6 banelings |
| 8 hydralisks & 2 banelings | 5 hydralisks & 5 banelings |
| 9 hydralisks & 1 baneling | 6 hydralisks & 4 banelings |
| 4 zerglings & 6 banelings | 7 hydralisks & 3 banelings |
| 6 zerglings & 4 banelings | 8 hydralisks & 2 banelings |
| 7 zerglings & 3 banelings | 9 hydralisks & 1 baneling |
| 8 zerglings & 2 banelings | 4 zerglings & 6 banelings |
| 10 zerglings | 5 zerglings & 5 banelings |
| 10 hydralisks | 6 zerglings & 4 banelings |
| 10 banelings | 7 zerglings & 3 banelings |
| 8 zerglings & 1 hydralisk & 1 baneling | 8 zerglings & 2 banelings |
| 1 zergling & 8 hydralisks & 1 baneling | 9 zerglings & 1 baneling |
| 7 zerglings & 2 hydralisks & 1 baneling | **Testing** |
| 2 zerglings & 7 hydralisks & 1 baneling | 10 zerglings |
| 5 zerglings & 3 hydralisks & 2 banelings | 10 hydralisks |
| 3 zerglings & 5 hydralisks & 2 banelings | 10 banelings |
| 4 zerglings & 4 hydralisks & 2 banelings | 8 zerglings & 1 hydralisk & 1 baneling |
| 3 zerglings & 4 hydralisks & 3 banelings | 1 zergling & 8 hydralisks & 1 baneling |
| **Testing** | 7 zerglings & 2 hydralisks & 1 baneling |
| 3 zerglings & 7 hydralisks | 2 zerglings & 7 hydralisks & 1 baneling |
| 8 zerglings & 2 hydralisks | 6 zerglings & 2 hydralisks & 2 banelings |
| 7 hydralisks & 3 banelings | 2 zerglings & 6 hydralisks & 2 banelings |
| 5 zerglings & 5 banelings | 5 zerglings & 3 hydralisks & 2 banelings |
| 9 zerglings & 1 baneling | 3 zerglings & 5 hydralisks & 2 banelings |
| 6 zerglings & 2 hydralisks & 2 banelings | 4 zerglings & 4 hydralisks & 2 banelings |
| 4 zerglings & 3 hydralisks & 3 banelings | 4 zerglings & 3 hydralisks & 3 banelings |
| 2 zerglings & 6 hydralisks & 2 banelings | 3 zerglings & 4 hydralisks & 3 banelings |

Table 3: Team formations in `Protoss` tasks

| 10_Protoss | 10_Protoss_Hard |
|---|---|
| **Training** | **Training** |
| 1 stalker & 9 zealots | 1 stalker & 9 zealots |
| 3 stalkers & 7 zealots | 2 stalkers & 8 zealots |
| 4 stalkers & 6 zealots | 3 stalkers & 7 zealots |
| 5 stalkers & 5 zealots | 4 stalkers & 6 zealots |
| 6 stalkers & 4 zealots | 5 stalkers & 5 zealots |
| 8 stalkers & 2 zealots | 6 stalkers & 4 zealots |
| 9 stalkers & 1 zealot | 7 stalkers & 3 zealots |
| 4 zealots & 6 colossi | 8 stalkers & 2 zealots |
| 5 zealots & 5 colossi | 9 stalkers & 1 zealot |
| 7 zealots & 3 colossi | 4 zealots & 6 colossi |
| 8 zealots & 2 colossi | 5 zealots & 5 colossi |
| 9 zealots & 1 colossus | 6 zealots & 4 colossi |
| 4 stalkers & 6 colossi | 7 zealots & 3 colossi |
| 5 stalkers & 5 colossi | 8 zealots & 2 colossi |
| 7 stalkers & 3 colossi | 9 zealots & 1 colossus |
| 8 stalkers & 2 colossi | 4 stalkers & 6 colossi |
| 10 stalkers | 5 stalkers & 5 colossi |
| 10 zealots | 6 stalkers & 4 colossi |
| 10 colossi | 7 stalkers & 3 colossi |
| 8 stalkers & 1 zealot & 1 colossus | 8 stalkers & 2 colossi |
| 1 stalker & 8 zealots & 1 colossus | 9 stalkers & 1 colossus |
| 2 stalkers & 7 zealots & 1 colossus | **Testing** |
| 6 stalkers & 2 zealots & 2 colossi | 10 stalkers |
| 5 stalkers & 3 zealots & 2 colossi | 10 zealots |
| 3 stalkers & 5 zealots & 2 colossi | 10 colossi |
| 4 stalkers & 4 zealots & 2 colossi | 8 stalkers & 1 zealot & 1 colossus |
| 4 stalkers & 3 zealots & 3 colossi | 1 stalker & 8 zealots & 1 colossus |
| **Testing** | 7 stalkers & 2 zealots & 1 colossus |
| 2 stalkers & 8 zealots | 2 stalkers & 7 zealots & 1 colossus |
| 7 stalkers & 3 zealots | 6 stalkers & 2 zealots & 2 colossi |
| 6 zealots & 4 colossi | 2 stalkers & 6 zealots & 2 colossi |
| 6 stalkers & 4 colossi | 5 stalkers & 3 zealots & 2 colossi |
| 9 stalkers & 1 colossus | 3 stalkers & 5 zealots & 2 colossi |
| 7 stalkers & 2 zealots & 1 colossus | 4 stalkers & 4 zealots & 2 colossi |
| 3 stalkers & 4 zealots & 3 colossi | 4 stalkers & 3 zealots & 3 colossi |
| 2 stalkers & 6 zealots & 2 colossi | 3 stalkers & 4 zealots & 3 colossi |

## B.2 Architecture, Training and Evaluation

The evaluation procedure is similar to the one in (Rashid et al., 2020). The training is paused after every 30k timesteps during which 16 test episodes are run with agents performing action selection greedily in a decentralised fashion. The percentage of episodes where the agents defeat all enemy units within the permitted time limit is referred to as the test win rate.

To speed up the learning, the agent networks are parameters are shared across all agents. A one-hot encoding of the `agent_id` is concatenated onto each agent's observations. All neural networks are trained using RMSprop without weight decay or momentum.

**Value-based baselines**

The architecture of all agent networks is a DRQN (Hausknecht & Stone, 2015) with a recurrent layer comprised of a GRU with a 64-dimensional hidden state, with a fully-connected layer before and after. We sample batches of 32 episodes uniformly from the replay buffer, and train on fully unrolled episodes, performing a single gradient descent step after 8 episodes.

Table 4: Hyperparameters of QMIX and VDN

| Method | Name | Value |
|---|---|---|
| QMIX & VDN | learning rate | $5 \times 10^{-4}$ |
| | RMSprop $\alpha$ | 0.99 |
| | replay buffer size | 5000 episodes |
| | target network update interval | 200 episodes |
| | $\gamma$ | 0.99 |
| | double DQN target | True |
| | initial $\epsilon$ | 1 |
| | final $\epsilon$ | 0.05 |
| | $\epsilon$ anneal period | 50000 steps |
| | $\epsilon$ anneal rule | linear |
| QMIX | mixing network hidden layers | 1 |
| | mixing network hidden layer units | 32 |
| | mixing network non-linearity | ELU |
| | hypernetwork hidden layers | 2 |
| | hypernetwork hidden layer units | 64 |
| | hypernetwork non-linearity | ReLU |

**PPO baselines**

We parameterize the actor and critic with two independent recurrent neural networks, each of which is comprised of a GRU with a 64-dimensional hidden state, with a fully-connected layer as the input and output.

Table 5: Hyperparameters of IPPO and MAPPO

| Method | Name | Value |
|---|---|---|
| IPPO & MAPPO | critic learning rate | 0.001 |
| | actor learning rate | 0.99 |
| | $\gamma$ | 0.99 |
| | $\lambda$ | 0.95 |
| | $\epsilon$ | 0.2 |
| | clip range | 0.1 |
| | normalize advantage | True |
| | normalize inputs | True |
| | grad norm | 0.5 |
| | number of actors | 8 |
| | critic coefficient | 2 |
| | entropy coefficient | 0 |
| | mini epochs for actor update | 10 |
| | mini epochs for critic update | 10 |
| | mini batch size | 64 |

## C    Full StarCraft II Results

Complete results for StarCraft II are as shown in Fig. 6, Fig. 7, Fig. 8.

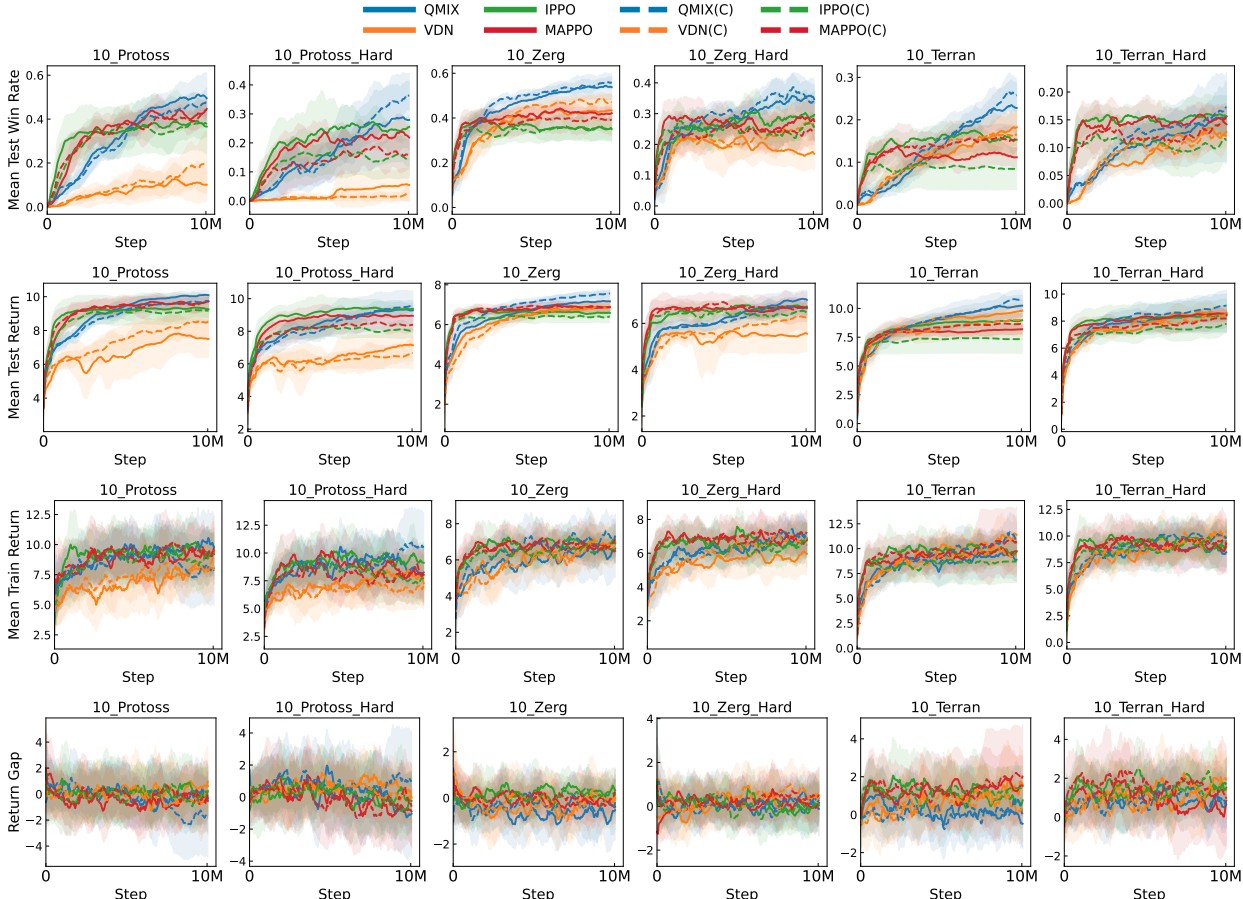

Figure 6: Experimental results on SMAC unit swapping tasks. Dashed lines indicate the inclusion of information on capabilities as part of the agent observations. Standard deviation is shaded.

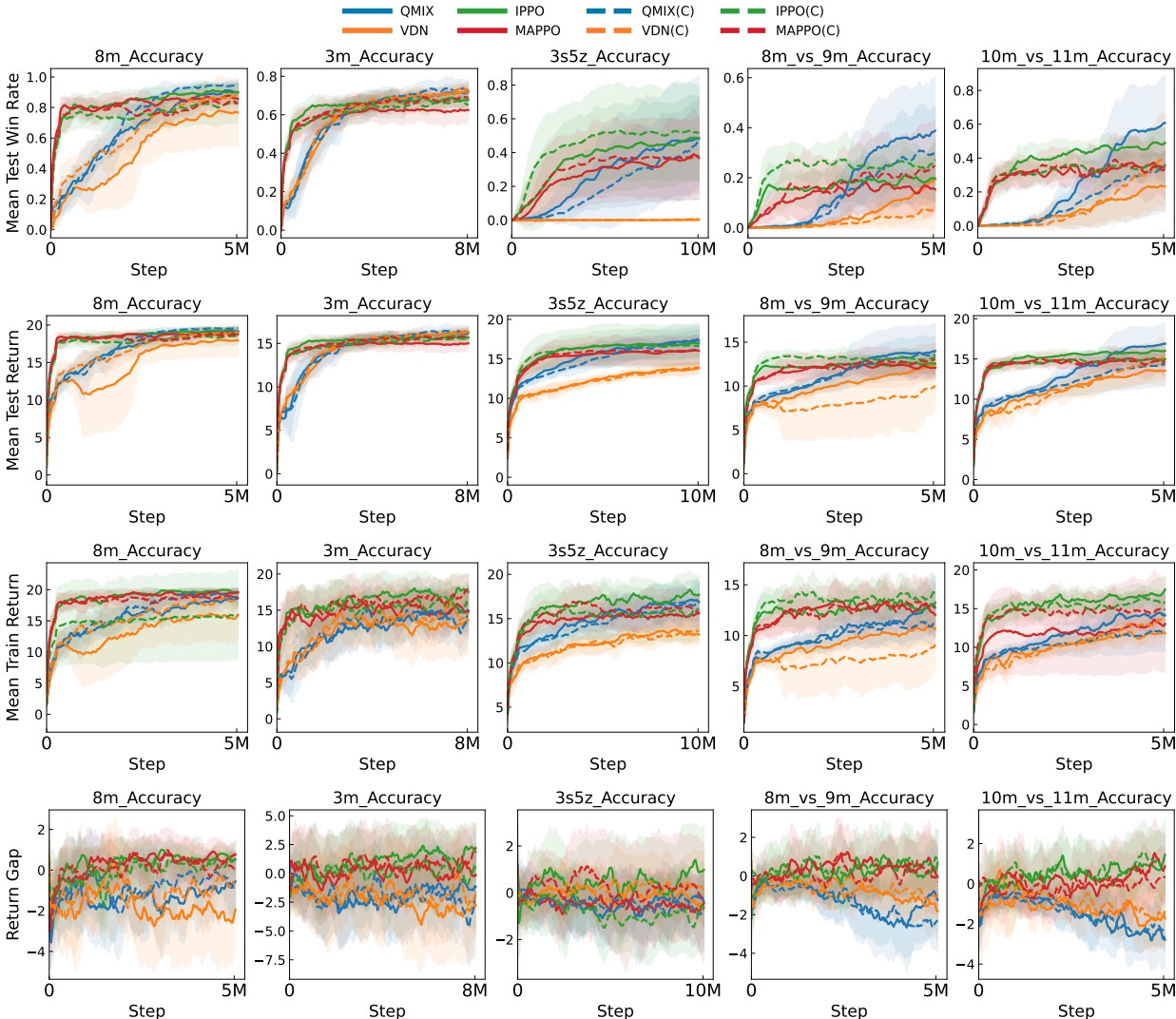

Figure 7: Experimental results on SMAC unit accuracy tasks. Dashed lines indicate the inclusion of information on capabilities as part of the agent observations. Standard deviation is shaded.

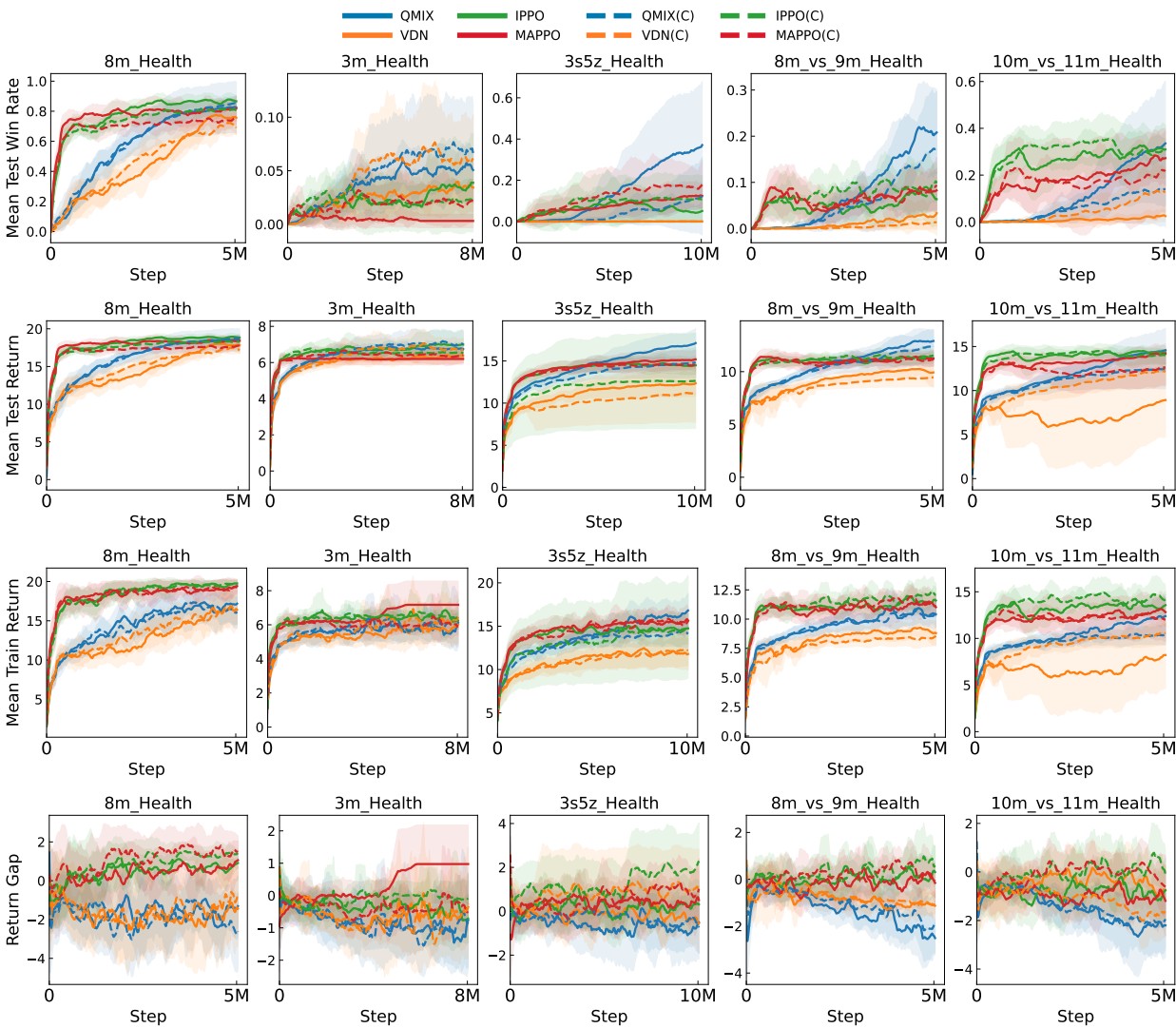

Figure 8: Experimental results on SMAC unit health tasks. Dashed lines indicate the inclusion of information on capabilities as part of the agent observations. Standard deviation is shaded.

