# OpenReview forum: "Generalization in Cooperative Multi-Agent Systems"
_TMLR — Rejected by TMLR_

### Review · Reviewer_aqGT · 2023-07-17

**Summary Of Contributions:**

This paper studies generalization in cooperative Multi-Agent Systems (MASs) from both the lens of theoretical analysis and empirical investigation. The authors focus on Combinatorial Generalization (CG), considering the generalization under flexible changes in terms of agent capacity, number  and etc.

For the theoretical aspect, the authors first make use of linear assumptions on reward and dynamics functions (with respect to agent capacities), and proposes generalization upper bounds for scenarios like transfer, substitution, population increase/decrease. Then the bounds are generalized to the cases when the assumptions are only approximately achieved, the capacities are inaccurately inferred and the rewards are Lipschitz continuous.

For the empirical investigation, the authors use typical MARL algorithms including QMIX, VDN, MAPPO, IPPO and Fruit Forage, Predator Prey and SMAC for examining the generalization bounds and CG performance of different algorithms.

**Audience:**

Yes

**Claims And Evidence:**

No

**Requested Changes:**

1. I think the limitations of (approximately) linear assumption should be discussed more, especially to use some problem instances to give more information on what kind of problems can be fine with linear assumption and when linear assumption will be improper. I think this is important to audiences.
2. More explanation and clarification should be added for the questions I raised above. Moreover, Figure 5 now is difficult to read in the sense that I cannot get the main message the authors want to convey (and the corresponding text should be polished since it helps little now).

**Strengths And Weaknesses:**

### Strengths:

- I appreciate the authors’ effort in studying CG in MARL, which I agree is very significant problem. The key properties listed in Section 1 and Figure 1 are clear.
- The generalization bounds of CG in MARL are proposed. Although the linear assumptions used seem to be limited, I think it is insightful and it is still a good step to theoretical underpinnings of CG. The connection between the linear assumption used in this paper and the SF makes it easy to understand the thoughts behind theoretical results presented.
- The experiments are conducted from different aspects.

&nbsp;

### Weaknesses:

- For the key properties P3, the authors mentioned that the agent needs to infer other agents’ capacity while this is not clear and seems to be neglected in the experiments (i.e., how agents infer the others’ capacity if capacities are unobservable). In addition, it seems that the key properties of nonstationary environment (P5) is not discussed and considered after the introduction.
- The modeling of agent capacity with (unobservable) vectors remind me some similar ways of modeling used by prior works in RL. Some related works are missing, e.g., opponent modeling for generalization[1,2], adhoc team[3], representation learning of agent policy[4].
- For the experiments, I think some results need further explanation:
    - In Figure 4b, I do not see a significant difference between solid and dashed lines (except for QMIX). Thus, I cannot understand the sentences in the first paragraph of Section 5.2 like “by the fact that knowing agent capabilities results in a lower generalization gap”.
    - How to explain the jump start of VDN/QMIX in Figure 4c?
    - How to understand that MAPPO and IPPO perform comparably in Figure 4 and 5?

&nbsp;

### Minor:

- The notations for expectation in this paper are inconsistent.

---
### Reference
[1] H. He and J. Boyd-Graber. **Opponent Modeling in Deep Reinforcement Learning.** ICML 2016

[2] D. Grover et al. **Learning Policy Representations in Multiagent Systems.** ICML 2018

[3] Gu et al. **Online Ad Hoc Teamwork under Partial Observability.** ICLR 2022

[4] Tang et al. **What about Inputting Policy in Value Function: Policy Representation and Policy-Extended Value Function Approximator.** AAAI 2022

---

> ### Author Response · Authors · 2023-08-20
>
> Thanks for the review, we are glad that you liked the novel problem of combinatorial generalization (CG) and the theoretical and empirical results we show for it in our work. We address your concerns below and have also updated the draft accordingly for them:
>
> >I think the limitations of (approximately) linear assumption should be discussed more, especially to use some problem instances to give more information on what kind of problems can be fine with linear assumption and when linear assumption will be improper. I think this is important to audiences.
>
> We wish to clarify that the analysis begins with the linear assumptions only for the ease of exposition (Theorems 1-5). **We discuss the most general case of arbitrary dependence** in Theorem 6 and Lemma 1 and provide insights into extending the rest of the results (Theorem 2-5) similarly. Note that **the non-linear dependence will always be Lipschitz bounded as long as the capabilities come from a bounded space**, thus Theorem 6 makes no restrictive assumptions. We have also added more insights into the arbitrary rewards case by formally introducing the $\alpha, k$-rewards in Sec 3 including discussion of why generalization becomes difficult in this regime (see Lemma 1). We also design the experiments covering these various settings Fruit Forage (linear), StarCraft, Predator-Prey (non-linear), see first para in Sec 4.1 in the updated draft.
>
> > Figure 5 implications
>
> The main conclusion in Fig 5 is that the generation to unseen tasks during test time becomes difficult as the "grounding" problem becomes difficult. We observe that agents are more likely to overfit instead of generalize in this domain, for instance see 8_m Accuracy where knowing agent capabilities information (dashed line) improves the test-time performance, but the corresponding generalization gap is worse thus implying over-fitting. We add more discussion on this in Sec 5.2 (para 5) in the updated draft.
>
> > For the key properties P3, the authors mentioned that the agent needs to infer other agents’ capacity while this is not clear and seems to be neglected in the experiments. In addition, it seems that the key properties of nonstationary environment (P5) is not discussed and considered after the introduction
>
> P3 is about grounding the capability and P5 is about inferring the capabilities in a non stationary environment, the reviewer seems to confuse between the two in their question. Note that since all the agents are learning and changing their policy during training, we are already dealing with non-stationarity. Basically, the oracle baseline (dashed plots) test for key properties P1-P4, whereas the solid line setting is more challenging as the capabilities of the teammates is not observed and it test all the key properties P1-P5.
>
> We also add more explanation about how we are testing for all the key properties P1-P5 in draft Sec 4.2.
> > Some related works are missing,
>
> Thanks, we have added them to the updated draft
>
> > I cannot understand the sentences in the first paragraph of Section 5.2 "This hypothesis is additionally supported by the fact that
> knowing agent capabilities results in a lower generalization gap"
>
> The statement is in context to agents successfully grounding the privileged information about teammate capabilities for the easier predator-prey task. The generalization gap is difference in performance between train and test tasks. It arises due to many factors including limitations of the algorithm itself like representation capacity etc. and ability to ground the capabilities. Ability to ground capabilities during train time doesn't necessarily imply the same would happen on unseen capabilities during test time. Whereas any difference in dashed vs solid plots for Generalization gap (here Fig 4 b) for a given algorithm is solely attributable to the fact that agents are able to better use the privileged information (ie. better grounding). We see this is true fro both QMIX and VDN.
>
> >Why MAPPO and IPPO perform comparably in Figure 4 and 5?
>
> IPPO can be quite effective, especially in scenarios where learning a good joint critic (MAPPO) is hard. Additionally, value based methods are more stable to changes in capability inputs as they are only taking argmax over the utility network outputs for selecting action, whereas for policy based methods, the network outputs can wildly change due to change in inputs.

---

### Review · Reviewer_dbg5 · 2023-07-30

**Summary Of Contributions:**

The paper presents generalization bounds in multi-agent reinforcement learning (MARL). The MARL setting is formulated as a contextual Markov decision process (MDP) where the context corresponds to the vector describing the agent capabilities. To obtain the generalization bounds, the authors assume that both the transition and reward functions are linear with respect to these vector capabilities. Various bounds are obtained with respect to changes of, e.g., team composition, capability, or team size. The authors also discusses an extension to the case where the assumption of linearity of the reward function is relaxed. Finally, experiments are conducted in three domains to evaluate the generalizability of various existing MARL algorithms.

**Audience:**

Yes

**Broader Impact Concerns:**

This is a theoretical paper, which I believe doesn't require such statement.

**Claims And Evidence:**

No

**Requested Changes:**

See the cons above.

Minor points:
"Combinatorial generalization" has been already used in machine learning. I suggest the authors to mention it.
- page 2, line 2: "the optimal policy": it may not be unique.
- page 3: V^\pi(s) was not defined in Section 2.
- (3): W_p doesn't fit the description below. Also, shouldn't it accept only 3 arguments and return a vector?
- page 4: "c_i^T" should be "c_i \cdot" to keep the notations consistent
- Theorem 1, line 2: "Then" -> "then"
In this theorem, the authors should mention earlier that y correspond to the old composition.


**Strengths And Weaknesses:**

Pros
As far as I know, the proposed MARL setting is novel.

The obtained bounds are generic and do not depend on algorithms.

Cons
I think the authors should discuss more the relation between their setting and contextual MDP or multi-task RL.

Regarding the bounds, since they are worst-case bounds, I am not sure how they are useful to help solve the generalization problem in MARL. The authors may want to comment on that point.

It is not clear to me that the assumptions made in the analysis hold in the experimental domains. I think the authors should discuss this important point.

The paper is generally clearly written, but there are a few issues (see below for some typos), especially in mathematical notations. Some notion like multiplexer policy  (Def. 1) is not defined. Also, is it guaranteed that such policy exists?

---

> ### Author Response · Authors · 2023-08-20
>
> Thanks for the review, we are glad that you liked the novel problem of combinatorial generalization (CG) and the theoretical and empirical results we show for it in our work. We address your concerns below:
>
> >Regarding the bounds, since they are worst-case bounds, I am not sure how they are useful to help solve the generalization problem in MARL. The authors may want to comment on that point.
>
> We have added a new section 5.3 for discussing how the analysis framework can be used to motivate a latent representation based approach towards better grounding and combinatorial generalization.
>
> >It is not clear to me that the assumptions made in the analysis hold in the experimental domains
>
> The analysis **starts with a relatively simple setting of linear dependence of rewards and transitions on capabilities in Eq 2,3** and **gradually discusses the most general case of arbitrary dependence**. Note that the dependence will always be Lipschitz bounded as long as the capabilities come from a bounded space, thus Theorem 6 makes **no restrictive assumptions**. We have also added more insights into the arbitrary rewards case by formally introducing the $\alpha,k$-rewards in Sec 3 including discussion of why generalization becomes difficult in this regime.
>
> **The motivation for the choice of the experimental domains is as follows**: The Fruit Forage
> follows the linear dependence in Eq. (2),Eq. (3) and is used to empirically validate the various bounds in
> Section 3 since the optimal policies can be manually computed for this domain. The Predator Prey and
> StarCraft II environments represent more challenging scenarios of non-linear dependence of the underlying
> reward and transitions on the agent capabilities discussed in Section 3 (Theorem 6, Lemma 1). This has also been added in Sec 4.1, first para in draft.
>
> Thus the experiments are designed to cover all the possibilities discussed in the Analysis section.
>
> >Multiplexer policy,  is it guaranteed that such policy exists?
>
> Multiplexer policy is only a theoretical concept to measure the regret against the best possible performance. We can always define it as a collection of optimal policies for each capability variation. Learning this policy is subject to practical constraints like number of training steps and model representation capacity.
>
> > $W_P$ doesn't fit the description below. Also, shouldn't it accept only 3 arguments and return a vector?
>
> $W_P$ takes 4 arguments and should be thought of as a tensor, missing argument implies function Currying. So $W_P$ returns a scalar when state, joint action, next state and the capability dimension index are all specified. For usage in Eq 3 as only state, joint action, next state are specified, the output is a d-dimensional vector over the free index(for capability dimension).
>
> >Discuss more the relation between their setting and contextual MDP or multi-task RL.
>
> We have added this in related works. Relation to contextual MDP in single agent RL is also mentioned in Sec 2
>
> **We have also modified the main draft for clarifying the above concerns.**

---

### Review · Reviewer_SkKo · 2023-08-06

**Summary Of Contributions:**

This paper looks at a very interesting problem of combinatorial generalization in cooperative multi-agent systems. When the problem admits certain linearity assumptions, value loss due to changes in capabilities and team compositions can be bounded. The theoretical bounds are validated empirically and incorporating capability as context sometimes leads to more sample efficient RL.

**Audience:**

Yes

**Claims And Evidence:**

Yes

**Requested Changes:**

The authors refer to "grounding" issues, speculating that when agents with context do better than agents without, they must be grounding better. But this claim seems largely unsubstantiated -- can you empirically verify that the difference in performance between PPO and value-based methods is because of grounding? Can you do the same to explain why adding context doesn't help in Figs 4 c,d?

**Strengths And Weaknesses:**

Strengths:
- Good introduction to the problem of combinatorial generalization
- Good balance between theoretical and empirical results
- The empirical evaluation of the theorems was convincing

Weaknesses:
- From what I understood, each agent gets to fully observe the capabilities of the other agents, which seems unrealistic.
- The RL results in harder tasks seem mostly inconclusive.

---

> ### Author Response · Authors · 2023-08-20
>
> Thanks for the review, we are glad that you liked the novel problem of combinatorial generalization (CG) and the theoretical and empirical results we show for it in our work. We address your concerns below:
>
> >From what I understood, each agent gets to fully observe the capabilities of the other agents, which seems unrealistic.
>
> We wish to clarify that **we do not propose a new method in the experiments**, instead we study how the combinatorial generalization (CG) behavior changes across the algorithms tested as we vary factors like amount of agent information and task hardness. The dashed line plots in which the agent fully observe the capabilities of the other agents is just **the best possible scenario baseline**, whereas the solid lines are one in which the teammate capabilities are not observed. We added more discussion about this in Sec 4.2.
>
> >The RL results in harder tasks seem mostly inconclusive
>
> From, the experiments, we conclude that the biggest challenge in attaining CG in complex domains is that of making the agents understand how their capabilities affect the team returns, which we refer as the **grounding** problem. We found that even the most informative baselines do not generalize well across the algorithms used for baselines, so creating such algorithms for this is an open problem which we motivate for future work. We have added more discussion on this in Sec 5 (esp Sec 5.3)
>
> > The authors refer to "grounding" issues, speculating that when agents with context do better than agents without, they must be grounding better. But this claim seems largely unsubstantiated -- can you empirically verify that the difference in performance between PPO and value-based methods is because of grounding?
>
> The only difference between dashed and solid line for any given algorithm is observing the teammate capabilities, other details like network architecture etc. is identical. Therefore, the performance difference between the two versions keeping the algorithm fixed is solely attributable to access to teammate capabilities. The only way to use capabilities is to "ground" them ie. meaningfully interpret them through the agent neural net. Better grounding is the only explanation if we see a lower generalization gap between train and test (unseen) tasks for dashed line.
>
> Note that **we are not using the above reasoning to compare between two different algorithms** like PPO and QMIX. The difference in performance across algorithms comes from a variety of reasons based on the algorithmic choices (like value based vs policy based), type of environment, sample complexity, behavior to noise etc. For instance, the difference in PPO vs Value based methods is likely because value based methods are more stable to changes in capability inputs as they are only taking argmax over the utility network outputs to act, whereas for policy based methods, the network outputs can wildly change due to change in inputs at test time.
>
> >Can you do the same to explain why adding context doesn't help in Figs 4 c,d?
>
> In scenario 4c,d the task is harder with both unseen team composition and unseen agent types. Here, when the oracle agents (dashed line) are provided the teammate capabilities, they are unable to ground it in the given training budget thus the capabilities act as noise for the agent neural networks deteriorating the performance.
>
> **We have also modified Sec 4,5 for clarifying the above concerns.**

---

### Author Response · Authors · 2023-08-20
**Draft Update**

We thank all the reviewers for their feedback.

We have updated the draft to address the concerns and requested changes from the reviewers, the edited portions are marked red.

We individually respond to the reviewer specific comments in the threads.

---

### Decision · Action_Editors · 2023-10-13

**Recommendation:** Reject

**Comment:**

During the discussion, two reviewers were "leaning reject" while the other recommended accepting the paper.

One reviewer leaning toward reject suggested "Given the theoretical results, it would have been more interesting to perform experiments with a latent space (as now suggested in Section 5.3) and verify the consequences of those results. Currently, apart from Section 5.1, the experiments consider the case where the agents observes the agent capabilities, which is not covered by the theoretical part."

The other reviewer leaning toward reject expressed concerns that some of the empirical results remained unconvincing due to the overlapped error bars in many of the plots.

Due to the extent of the potential revisions as suggested by the reviewers (reducing the disconnect between the theoretical and empirical aspects, and additional runs/revision of plot error bars, fixing notation issues), this article is not yet ready for publication.

However, based on its strengths as listed above and in the individual reviews, **I want to strongly encourage the authors to revise and resubmit the article to TMLR**. I do believe that this would be an appropriate venue for this work, and that with these revisions it would be a worthwhile addition to the literature.

**Audience:**

This work on MARL would be of interest to several communities, including researchers that work on multi-agent systems and those that study reinforcement learning. Additionally, the combination of theoretical analysis and empirical study makes it appealing to a wide audience.

**Claims And Evidence:**

This paper explores generalization in cooperative multi-agent reinforcement learning (MARL), first presenting a theoretical analysis yielding generalization bounds, and then examining generalization in a variety of current MARL algorithms. The problem formulation and investigation are novel and intriguing, the theoretical results are significant, and the authors include extensive supporting empirical results on current methods. In particular, the paper does a very nice job exploring the different dimensions (e.g., team size, composition, etc.) of the MARL problem and how they affect generalization, albeit with some reasonable assumptions of linearity (and relaxation of this assumption) to make the analysis feasible. In their most recent revision, the authors addressed many of the concerns raised by the reviewers in the initial review.

However, during the discussion, there were shared concerns among two reviewers that some issues remained with the empirical evaluation that were not adequately addressed. In particular, that there was a disconnect between the theoretical analysis and the empirical evaluation, and that some of the results remained unconvincing, both as described further below.

Additionally, the reviewers pointed out that the revised draft still contained many typos and notation issues that needed to be corrected.

**Resubmission Of Major Revision:**

The authors may consider submitting a major revision at a later time.